# Survival of the Qaidam Mega-Lake System under Mid-Pliocene Climates and its Restoration under Future Climates

Dieter Scherer[1]

[1] Chair of Climatology, Technische Universität Berlin, Berlin, 12165, Germany

*Correspondence to*: Dieter Scherer (dieter.scherer@tu-berlin.de)

**Abstract.** The Qaidam Basin in the north of the Tibetan Plateau has undergone drastic environmental changes during the last millions of years. During the Pliocene, the Qaidam Basin contained a freshwater mega-lake system although the surrounding regions showed increasingly arid climates. With the onset of the Pleistocene glaciations, lakes began to shrink, and finally disappeared almost completely. Today, hyperarid climate conditions prevail in the low-altitude parts of the Qaidam Basin. The

question, how the mega-lake system was able to withstand the regional trend of aridification for millions of years, remained enigmatic, so far. This study reveals that the mean annual water balance, i.e., the mean annual change in terrestrial water storage in the Qaidam Basin, is nearly zero under present climate conditions due to positive values of net precipitation in the high mountain ranges, and shows positive annual values during warmer, less dry years. This finding provides a physically based explanation, how mid-Pliocene climates could have sustained the mega-lake system, and that near-future climates not

much different from present conditions could cause water storage in reservoirs, rising lake levels and expanding lake areas, and may even result in restoration of the mega-lake system over geological time scales. The study reveals that a region discussed as being an analogue to Mars due to its hyperarid environments is at a threshold under present climate conditions, and may switch from negative values of long-term mean annual water balance that have prevailed during the last 2.6 million years to positive ones in the near future.

## 1 Introduction

Paleogeographic studies (Chen and Bowler, 1986; Huang et al, 1993; Mischke et al., 2010; Wang et al., 2012, Fang et al., 2016) on the intermontane endorheic Qaidam Basin (QB), located in China's desert region in the north of the Tibetan Plateau (TP), revealed that it once contained a mega-lake system during the mid-Pliocene (ca. 3.3-3.0 Ma BP) and before. Although spatial details of the mega-lake system are not known, Chen and Bowler (1986) have reported that total lake surface was about

59.000 km$^2$ during the early Pleistocene. The onset of the Pleistocene glaciations at ca. 2.6 Ma BP marked a period of increased variability of climate, lake level and extent, as well as changes in salinity (Huang et al, 1993; Wang et al., 2012; An et al., 2001; Heermance et al., 2013, Fang et al., 2016). The mega-lake system finally disappeared during the last 100 ka (Madsen et al., 2014; Yu et al., 2019). Today, only few saline lakes and playas exist, and the low-altitude parts of the QB are hyperarid deserts (Chen and Bowler, 1986; Huang et al, 1993; Wang et al., 2012).

Paleoclimate studies (e.g. An et al., 2001; Fang et al., 2007, 2016; Miao et al., 2013; Koutsodendris et al., 2019) could prove that climates in the region have become increasingly dry throughout the Pliocene. However, the existence of a mega-lake system during this period implies that long-term mean annual water balance $\Delta S$, i.e., the total annual change in terrestrial water storage within the basin's reservoirs (aquifers, soils, lakes, rivers, permafrost, snow covers, glaciers, etc.), was close to zero, and did not show, on average, negative values over time periods of thousands of years or longer, because otherwise, the mega-

lake system would have temporarily dried out and produced layers of evaporites.

The water balance equation of a drainage basin can be written as

$$\Delta S = P - ET - R \tag{1}$$

where $P$ is precipitation and $ET$ is actual evapotranspiration, both quantities spatially averaged over the area of the drainage basin. Total runoff is indicated by $R$, which is the sum of surface and groundwater runoff leaving the drainage basin. In this

study, all quantities in Eq. (1) are expressed as volume water equivalent per area and time interval (in mm/month or mm/a). For endorheic basins like the QB surface runoff is zero by definition. Groundwater entering or leaving a basin is generally difficult to quantify but can be neglected for the water balance of large intermontane basins like the QB, which has an area of ca. 254,000 km². Thus, the change in terrestrial water storage $\Delta S$ of the QB can be approximated by

$$\Delta S = P - ET \tag{2}$$

The term $P - ET$ is, as in this study, often referred to as net precipitation (Morrow et al., 2011), and sometimes called effective precipitation (Pritchard et al., 2019) or water availability (Greve et al., 2018). In this study, $\Delta S$ is called water balance and refers to the spatial average of net precipitation over the total area of the QB by applying Eq. (2).

This study addresses the research question how mean annual water balance in the QB could have been zero or even positive over millions of years under very dry climates such that a mega-lake system could have been sustained. I hypothesise that the

high mountain ranges in the QB receive sufficient precipitation such that negative values of net precipitation at lower altitudes due to very low amounts of precipitation are compensated by positive values of net precipitation at high altitudes. A second hypothesis that is tested in this study is that annual $\Delta S$ of the QB is positively sensitive to warmer climates, i.e., that positive anomalies in spatially averaged annual air temperature $T$ cause positive anomalies in annual $\Delta S$ under present geographic conditions. Knowing the value of the sensitivity $\frac{\partial \Delta S}{\partial T}$ of annual $\Delta S$ to changes in annual $T$ would allow for a first-order estimate

of the water balance under different climates like those in the mid-Pliocene or the future. This approach follows the general idea underlying studies that have linked climates of the past with those projected for the near future using the concept of climate sensitivity (e.g. Chandar and Peltier, 2018).

This study does, however, neither intend to reconstruct the climates or the hydrology of the past nor to make predictions for the future. So far, paleogeographic data like paleo-altitudes or paleo-hydrographic conditions of the QB, or past large-scale

atmospheric circulation patterns in High Mountain Asia are not well constrained by observational data, making it difficult to use them as input in spatially detailed hydrological models. It is also difficult to quantify feedbacks in the hydrological cycle of the QB like an increase of $ET$ due to increased lake area, changes in blocking of air masses under different orography, or

changes in water recycling within the QB under more humid atmospheric conditions. Moreover, a physically based explanation for the proven long-term existence of the mega-lake system in the QB under dry climate conditions is sought.

Since the mid-Pliocene is often regarded as past analogue for modern climate changes (Zubakov and Borzenkova, 1988; Haywood et al., 2016; Chandar and Peltier, 2018), this study also intends to provide a rational basis for assessing environmental changes that might be caused by climate changes as projected in this region for the future (Burke et al., 2018; Gu et al., 2018; Hui et al., 2018).

## 2 Data and methods

Fig. 1 provides an overview on the QB and its surrounding regions in the north of the TP. The boundary of the QB was delineated by Lehner and Grill (2013) from a digital elevation model (DEM) derived from data of the Shuttle Radar Topography Mission (SRTM).

### 2.1 Meteorological data from the High Asia Refined analysis (HAR)

Meteorological data for 14 hydrological years (2001-2014) covering the period from 10/2000 to 09/2014 were taken from the
first version of the High Asia Refined analysis (HAR) data set (Maussion et al., 2011, 2014). In the study region, hydrological years start in October, and are numbered by the calendar year to which January belongs (i.e., the hydrological year 2001 starts in October 2000). The HAR data set was produced by dynamical downscaling of global gridded atmospheric data of the National Centers for Environmental Prediction (NCEP) Operational Model Global Tropospheric Analyses (FNL) with the Weather Research and Forecasting (WRF) model version 3.3.1 to two regional domains of 30 km and 10 km grid spacing as
described by Maussion et al. (2011, 2014). In contrast to regional climate simulations, the WRF model is re-initialised every day, and integrated only over 36 hours. Data of the first twelve hours are discarded to avoid artefacts eventually caused by model spin-up. The resulting data covering a full day are thus strongly constrained by the observed large-scale state of the atmosphere. Temporal resolution of the HAR 10 km data set is one hour (three hours for the HAR 30 km data set). HAR data used in this study were aggregated to monthly values. Monthly data for air temperature $T$ (2 m above ground) and specific
humidity $q$ (2 m above ground), which are called climate drivers in this study, are monthly means, while monthly data of $P$, which comprises both rainfall $P_{rain}$ and snowfall $P_{snow}$, and of $ET$ are monthly sums. Monthly values for $\Delta S$ were computed from monthly values of $P$ and $ET$ spatially averaged over the QB by applying Eq. (2). Data were further aggregated to annual means and sums for each hydrological year, and to mean monthly and annual values for the 14-year study period, both for each of the 2543 grid points of the HAR 10 km data set (280 grid points in the HAR 30 km data set) and the spatial averages
over the QB.

**2.2 Meteorological data from the Global Summary of the Day**

The National Centers for Environmental Information (formerly known as National Climatic Data Center) of the National Oceanic and Atmospheric Administration (NOAA) provide the Global Summary of the Day (GSOD), which is a collection of meteorological data from numerous weather stations all over the world. Eight GSOD stations providing meteorological data for the entire study period are located within or nearby the QB as shown in Fig. 1. Only GSOD data that are not flagged as invalid were used in this study. The data set is used for a discussion of error and uncertainties (see Section 4).

**2.3 Data on actual evapotranspiration**

A data set of annual $ET$ values for the QB and eight hydrological subregions covering the time period from 2001 to 2011 (Jin et al., 2013) was used in this study for assessing the results based on the HAR 10 km data set. The data set was derived from various data sources, especially from space-borne remote sensing data from the Moderate Resolution Imaging Spectroradiometer MODIS, by applying the Surface Energy Balance System (SEBS) algorithm (Su, 2002). The data set is used for a discussion of error and uncertainties (see Section 4).

**2.4 Statistics**

Regression analyses as performed in this study are simple linear regressions and one multiple linear regression using the ordinary least squares method. Simple linear regressions, in which altitude $h$ was used to as predictor variable $x$, were performed both by using $x = h$ directly (in m a.s.l.), and by $x = e^{h^*}$, where $h^* = \frac{h}{h_{scl}}$ ($h_{scl} = 1000$ m). The respective result with the highest effect size is used in this study. Probability values $p$ for significance testing of the regression results were computed from double-sided $t$-tests. Statistically significant results are assumed for $p < 0.05$. The effect size of the regressions is specified by the coefficient of determination $r^2$, which is the fraction of variance in the dependent variable $y$ that is explained by the linear model of the predictor variable(s) $x$. The effect size is further specified by the adjusted $r^2$ ($r^2_{adj}$):

$$r^2_{adj} = 1 - (1 - r^2)\frac{N-1}{N-k-1} \qquad \text{Eq. (3)}$$

where $N$ is the number of observations (i.e., years; $N = 14$), while $k$ is the number of predictor variables.

**3 Results**

In the following subsections, the study results are presented. First, annual values of the water balance of the QB are shown for the 14 years. Then, altitude dependencies of each quantity are presented for each grid point of the HAR 10 km data set within the QB. Finally, the results of a sensitivity study relating the water balance components with the climate conditions are shown.

## 3.1 Water balance

Table 1 lists mean monthly and annual values for the water balance components and the climate drivers, while Fig. 2 presents time series of the annual values of $P$, $ET$, and $\Delta S$ for the 14 years covered by this study. The results show that the QB's water balance is nearly zero ($\Delta S$ = -14±34 mm/a) under present climate conditions. From 2005 to 2012, all years except 2011 show above-average annual values for the water balance due to increased annual precipitation. Rainfall is the main driver of interannual variability of both precipitation and water balance, while snowfall and actual evapotranspiration are less variable. The first year (2001) was the one with the by far most negative water balance ($\Delta S$ = -94 mm/a), and was also the coldest ($T$ = -1.6 °C) and driest ($q$ = 2.2 g/kg) year, and received the lowest amount of precipitation ($P$ = 122 mm/a), rainfall ($P_{rain}$ = 45 mm/a), and snowfall ($P_{snow}$ = 77 mm/a) (see Tables S1-S7 in the Supplement for monthly and annual values for each quantity and year).

Mean monthly values for the quantities shown in Table 1 illustrate strong differences in their seasonality. While $T$, $q$, $P$, $P_{rain}$, $P_{snow}$, and to a lesser extend also $ET$, show strong variations over the year with slightly displaced winter minima and summer maxima, $\Delta S$ is less variable during the course of the year. This indicates complex interdependencies between the climate drivers and the water balance components, which lead to partial compensatory effects. Concurrence of the summer maxima of air temperature and precipitation in July leads to a shift of the maximum of snowfall to May, since higher air temperatures during summer reduce the fraction of precipitation falling as snow. The cold months are connected with slightly negative monthly water balance due to sublimation of snow exceeding snowfall. The concurrently higher values of precipitation and actual evapotranspiration during summer mostly compensate each other such that monthly water balance is only slightly positive during early summer. In August monthly water balance even shows small water losses since precipitation decreases faster than actual evapotranspiration.

## 3.2 Altitude dependencies

Altitude dependencies of the annual water balance components and the climate drivers are presented in Fig. 3. While $T$, $P$, $P_{snow}$, and $P - ET$ show strong correlations with altitude $h$ within the QB, $ET$ is only weekly correlated with $h$, and $q$ is not dependent on $h$. At altitudes of 4000 m a.s.l. and higher, net precipitation becomes positive on average, and none of the HAR 10 km grid points shows negative values above 4700 m a.s.l., which demonstrates the importance of high mountains as water towers (Xu et al., 2008) for the hydrology of the QB, especially under arid climates like those in the desert zones of High Mountain Asia.

Fig. 3 illustrates that the air over the lakes (ten grid points in the HAR 10 km data set) is generally warmer and less dry than over land, which is the result of high actual evapotranspiration due to lake evaporation. Mean annual values for $T$, $q$, $P$, $P_{snow}$, $ET$, and $P - ET$ averaged over the ten lake grid points are 0.7 °C, 4.6 g/kg, 153 mm/a, 60 mm/a, 649 mm/a, and -496 mm/a, while the respective values for the land grid points are -0.5 °C, 2.6 g/kg, 201 mm/a, 103 mm/a, 212 mm/a, and -12 mm/a.

Although there are a few low-lying areas with very high actual evapotranspiration, the majority of land areas shows increasing actual evapotranspiration with altitude, while air temperature strongly decreases with altitude (following the mean annual moist-adiabatic lapse rate). This indicates that abundance of water but not available energy for latent heat is the main limiting factor for actual evapotranspiration in the QB. In fact, the areas showing high actual evapotranspiration are mainly saline lakes that are not indicated as lakes in the HAR 10 km data set, such that water is available for actual evapotranspiration.

Lakes tend to suppress rain- and snowfall such that their annual values are lower over lakes. In consequence, most lakes (seven of ten grid points) show strongly negative net precipitation. Further details and maps of the spatial patterns of the water balance components and the climate drivers are presented in Fig. S1 to S7 in the Supplement.

## 3.3 Sensitivity study

Fig. 4 presents results of simple linear regression analyses between annual values of the spatially averaged quantities for the entire QB for the 14 hydrological years, which reveal that annual variability of water balance is driven by precipitation but not by actual evapotranspiration. Both precipitation and water balance are themselves driven by air temperature and specific humidity, the latter showing much stronger effect sizes in the simple linear regressions.

Since annual air temperature and specific humidity are themselves correlated ($r^2 = 0.571$; $r_{adj}^2 = 0.535$; $p < 0.01$; see Fig. S8 in the Supplement), the problem of multicollinearity of the two climate drivers was addressed in this study by an additional multiple linear regression, in which both annual $T$ and $q$ serve as predictor variables, while annual $\Delta S$ is taken as the dependent variable. Both predictors together explain more than 85 % of the variance in annual $\Delta S$ ($r^2 = 0.852$; $r_{adj}^2 = 0.825$; $p < 0.001$). However, the analysis revealed that the unique contribution of annual air temperature to the variance in the annual water balance is not significant. Air temperature alone uniquely explains only 0.25 % ($p_T = 0.67$) of the variance in water balance, while the unique variance explanation by specific humidity is 31.96 % ($p_q < 0.001$). The combined effect of air temperature and specific humidity explains 52.97 %. Thus, the simple linear regression between specific humidity and water balance as shown in Fig. 4 (lower right panel) accounts for both direct and indirect effects of variations in annual specific humidity on annual water balance of the QB.

The results show that a change in annual specific humidity of 1.0 g/kg would lead to an estimated change in annual water balance of 131 mm/a. Since the standard error of the estimate for annual water balance is 14.0 mm/a, even a slight change in annual specific humidity of e.g. 0.2 g/kg would have a strong, significant effect such that annual water balance of the QB would become positive.

## 4 Discussion

### 4.1 Errors and uncertainties

Physical consistency of the results obtained from the HAR data set is ensured by the fact that the WRF model is physically based, and HAR data have been comprehensively analysed, particularly with respect to precipitation and atmospheric water transport (Maussion et al., 2011, 2014; Curio et al., 2015; Pritchard et al., 2019; Yoon et al., 2019, Bai et al., 2020). HAR data have been successfully utilised for e.g. studying glacier mass balance on the TP (Mölg et al., 2014), in which independent data sets from global reanalysis data and field measurements have also been included to ensure validity of the results. Pritchard et al. (2019) showed for the upper Indus basin that the HAR 10 km data set is particularly applicable in studies on water availability. Li et al. (2020) state that "…the two HAR data sets performed better at capturing the average precipitation in high-altitude areas than the other seven data sets…".

Since the HAR data set, as any data set, comes with errors and uncertainties, the question arises how they would influence the results of the study. Fig. 3 illustrates that at six of the eight GSOD stations within or nearby the QB the HAR 10 km precipitation data are well according to the measurements, while precipitation might be slightly underestimated at two GSOD stations by the HAR 10 km data. The altitudinal changes of both air temperature and precipitation as documented by the GSOD data are well captured by the HAR 10 km. A comparison of HAR 10 km results for actual evapotranspiration with those from the SEBS-based study by Jin et al. (2013) indicates higher actual evapotranspiration in the HAR 10 km data. During the calendar years from 2005 to 2011, mean annual actual evapotranspiration is 218 mm/a in the HAR 10 km data, while SEBS data show 153 mm/a. Both annual time series are well correlated during this time period ($r^2 = 0.733$; $r^2_{adj} = 0.679$; $p < 0.05$; see Fig. S9 and Table S8 in the Supplement). SEBS shows inconsistent, even lower actual evapotranspiration values during the calendar years 2001 to 2004 (HAR 10 km: $ET$ = 202 mm/a; SEBS: $ET$ = 77 mm/a; see Table S8 in the Supplement). These findings reveal that there is no evidence that the HAR 10 km data set would overestimate annual water balance of the QB. On the contrary, if SEBS values for annual actual evapotranspiration would be considered to be more accurate than HAR 10 km data, then annual water balance would have been positive throughout all years.

### 4.2 Comparison with other studies

The results of this study are in accordance with results from other studies. A number of studies (Liu and Chen, 2000; Kang et al., 2010; Li et al., 2010; Zhang et al., 2013b) revealed that the TP experiences general, but spatially and temporally varying trends to higher air temperatures, increasing humidity, and precipitation, which are also found in the QB (Wang et al., 2014). The regions in the Qinghai province, to which the QB belongs, show trends to warmer and wetter climates while those regions belonging to Tibet tend to warmer but dryer climates (Zhang et al., 2013b). However, recent precipitation trends and subsequent environmental changes show complex spatial and temporal patterns, such that wetting is also taking place in several regions of Tibet (see e.g. Yang et al., 2014; Li et al., 2019b) including endorheic basins in interior Tibet.

Several studies (Zhang et al., 2011; Zhang et al., 2013a; Lei et al., 2014; Zhang et al., 2017; Li et al., 2019a, Li et al., 2019b) reported on rising lake levels and expanding lake areas on the TP. Besides enhanced glacier melt, increasing precipitation is regarded as main driver of rising lake levels (Zhang et al., 2013a; Lei et al., 2014; Zhang et al., 2017, Li et al., 2019b). The total lake area in the QB has increased from 994 km$^2$ in the 1960s to 1046 km$^2$ in 2014 (Wan et al., 2016), and the number of lakes has increased (Li et al., 2019a) by 18 from 1977 to 2015. These findings further support that HAR 10 km data do not overestimate annual water balance in the QB, because lake growth indicates positive water balance.

Increased terrestrial water storage in the QB does not only affect lakes but also groundwater reservoirs. Jiao et al. (2015) showed for the QB that aquifers were recharged between 2003 and 2012 due to changes in terrestrial water storage of 20.6 km$^3$, which is equivalent to a slightly positive mean annual water balance of 8 mm/a during this time period, while the mean value from the HAR 10 km data set for the same time period is only slightly lower and amounts to 0 mm/a. This result is also confirmed by the study of Loomis et al. (2019) who show zero to slightly positive mass trends in the northern TP.

### 4.3 Implications for the mid-Pliocene and the future

During the mid-Pliocene, climates have been generally warmer and less dry (or wetter) than today in many regions of the world (Haywood et al., 2013, 2016). The studies of Zhang et al. (2013c) and Mutz et al. (2018) provide quantitative estimates of mid-Pliocene changes in mean annual air temperature and precipitation with respect to preindustrial climates from various global paleoclimate simulations. The study of Zhang et al. (2013c) is based on a multi-model ensemble, which showed approx. 2 to 4 K higher mean annual air temperature in the QB during the mid-Pliocene as inferred from their Fig. 6. The same Fig. 6 indicates 100 to 300 mm/a higher values for mean annual precipitation. The study of Mutz et al. (2018) is based on simulations by a single global model, showing 2 to 6 K higher mean annual air temperature in the high mountains, while it was approx. 2 to 4 K cooler in the lower parts of the QB as inferred from their Fig. 4. Mean annual precipitation was approx. 100 to 300 mm/a higher as shown in the same Fig. 4. Both studies did not present data on differences between preindustrial and present times. Nevertheless, the values for mean annual air temperature and precipitation as presented by Fig. 4 in Mutz et al. (2018) are generally comparable to those from the HAR 10 km data set. Thus, present mean annual air temperature is assumed to be slightly higher (about 1 K) than during preindustrial times, such that changes in air temperature between the mid-Pliocene and present times are slightly lower but still positive (at least 1 K higher than today). Analogously, changes in precipitation are assumed to follow the same pattern (at least 50 mm/a higher than today).

Table 2 presents estimates for mean annual changes in the water balance of the QB due to climate changes with respect to present conditions. The first three rows (marked in red) indicate changes $\Delta q$, $\Delta P$, and $\Delta(\Delta S)$ for estimates of minimum and maximum changes of annual air temperature $\Delta T$. As conservative estimates, minimum and maximum changes in annual air temperature were set to 1 and 2 K, respectively, which can be used as estimates for the air temperature range representing both mid-Pliocene climates and those projected for the end of the 21$^{st}$ century (Burke et al., 2018; Gu et al., 2018; Hui et al., 2018). Applying the results from the simple linear regressions (Fig. 4) providing estimates for the sensitivities $\frac{\partial q}{\partial T}$, $\frac{\partial P}{\partial T}$, and $\frac{\partial \Delta S}{\partial T}$, the

estimated changes in annual precipitation due to changes in annual air temperature would be 52 to 105 mm/a, which are compatible with the values modelled for the mid-Pliocene by Zhang et al. (2013c) and by Mutz et al. (2018). The estimated change in annual water balance as inferred from the changes in mean annual air temperature would lead to a positive mean annual water balance of 49 mm/a. Based on the same estimates for changes in annual air temperature and the sensitivity of annual specific humidity to changes in annual air temperature, the resulting changes in annual specific humidity would be between 0.3 and 0.6 g/kg.

Applying the sensitivities of annual precipitation $\frac{\partial P}{\partial q}$ and water balance $\frac{\partial \Delta S}{\partial q}$ with respect to changes in annual specific humidity (fourth and fifth row in Table 2; marked in blue), the resulting changes in annual precipitation and water balance would be almost identical (values differ only by 3 mm/a or less) to those directly estimated from the changes in annual air temperature. The sixth row of Table 2 (marked in black) shows the results for changes in annual water balance $\Delta(\Delta S)$ for given changes in annual precipitation $\Delta P$. Minimum and maximum values for the changes in annual precipitation (50 to 100 mm/a) are conservative estimates but also compatible with the studies of Zhang et al. (2013c) and Mutz et al. (2018). The mean change in annual water balance as inferred from the changes in annual precipitation by applying the sensitivity of annual water balance to changes in annual precipitation $\frac{\partial \Delta S}{\partial P}$ would lead to a positive mean annual water balance of 40 mm/a.

Warmer and less dry conditions in the QB are also confirmed from geological evidence (Miao et al., 2013; Wu et al., 2011; Cai et al., 2012). The very high sensitivity of annual water balance to changes in annual air temperature and specific humidity as revealed in this study would explain that even slightly warmer and less dry climates could result in positive long-term mean annual water balance such that the mega-lake system could have been able to exist in this still very dry region for long time. It must, however, be noted that these estimates do not consider feedbacks that will have additional effects on the water balance of the QB. At a certain point, lake area will not be able to increase further since increasing lake evaporation would drop positive to zero mean annual water balance. This feedback mechanism was analysed in this study by a semi-empirical model (see R source code in the Supplement) to provide first-order estimates on the effects of changes in lake extent and mean annual precipitation in the QB on actual evapotranspiration, net precipitation, runoff from land areas to lakes, as well as water storage and associated lake-level changes in the QB. The model considers that an enlarged lake area would increase total ET in the QB, since lake evaporation is much higher than actual evapotranspiration over land. It further assumes that runoff generated from positive values of net precipitation over land areas is routed through rivers and aquifers to the lowest parts of the QB after groundwater reservoirs have been recharged, and accumulates there, thus forming new lakes. Then, lake level rises, lake extent and total lake evaporation increase, and water balance of the entire QB is reduced. The model can thus be applied to assess equilibrium lake states by inverse modelling of lake extent. Based on input data on lake extent, changes in mean annual precipitation in the QB with respect to present-day situation (200 mm/a), and mean annual rate of lake evaporation, it computes the resulting mean annual water balance for the QB and the mean annual change in lake level, which need both to be close to zero for equilibrium lake states (for more details see the well-documented R source code in the Supplement).

The analysis was performed for three different values for annual lake evaporation, i.e., 600, 800, and 1000 mm/a, using lower and upper estimates from the HAR 10 km data, which are also supported by other studies. Lazhu et al. (2016) report 832 mm/a annual lake evaporation for the Lake Nam Co, while Li et al. (2016) have measured 824-833 mm/a for the nearby Qinghai Lake. Mean annual lake evaporation in the QB is 649 mm/a according to the HAR 10 km data set. These results indicate that actual lake evaporation is much less than potential evaporation over lakes (Lazhu et al., 2016; Li et al., 2016).

Projections were also computed for two values of changes in mean annual precipitation of 50 and 100 mm/a. These values have been chosen following the results summarized in Table 2. Finally, the change in mean annual precipitation was computed for each of the three values of mean annual lake evaporation such that the maximum extent of the mega-lake system in the QB (see illustration in Fig. 5) of approx. 59.000 km$^2$ as reported by Chen and Bowler (1986) would be sustained.

   Table 3 summarizes the results from the analysis. The values for the approximate altitudes of lake levels for equilibrium lake

states are based on the HAR 10 km model topography (see also Fig. S10 in the Supplement illustrating different equilibrium lake stands for the results presented in Table 3). Assuming a slight increase in mean annual precipitation of 19 mm/a, which is within the error bounds of precipitation in the HAR 10 km data set, would be sufficient to sustain the present-day lakes in the QB. Even in the most extreme case for the mid-Pliocene, i.e., a rather low increase of mean annual precipitation of only 50 mm/a and a very high rate of mean annual lake evaporation of 1000 mm/a would result in a lake extent of 6654 km$^2$, which

is much larger than today's largest lake in China, the Qinghai Lake (about 4300 km$^2$). This would require a rise in lake level of only 19 m. Assuming a linear decrease of actual evapotranspiration over time, starting from 40 mm/a until dropping to 0 mm/a, the time required to reach the new equilibrium would be approximately 1000 years. On the other side, a change in mean annual precipitation of 100 mm/a, combined with a low rate of mean annual lake evaporation of 600 mm/a, would lead to an equilibrium lake state that is characterized by a lake extent of 25.773 km$^2$ and an associated rise in lake level of 58 m.

This would be truly a mega-lake system. Even a high rate of mean annual lake evaporation would result in a lake extent of more than 15.000 km$^2$.

   The change in mean annual precipitation required to sustain the maximum extent of the mega-lake system varies between 210 mm/a (for mean annual lake evaporation of 600 mm/a) to 330 mm/a (for mean annual lake evaporation of 1000 mm/a). These values are higher than the conservative estimates of this study (50-100 mm/a), but still within or nearby the bounds of

the studies by Zhang et al. (2013c) and Mutz et al. (2018). Only the results for 1000 mm/a as value for mean annual lake evaporation are slightly beyond the upper limits given by Zhang et al. (2013c) and Mutz et al. (2018) for changes in mean annual precipitation (300 mm/a). The maximum extent of the mega-lake system would require a rise in lake level of 133 m, and the approximate altitude of the mega-lake system would be 2786 m a.s.l., which is in accordance with the geological studies on highest lake levels in the QB. Chen and Bowler (1986) have reported an approximate altitude of the highest lake

level of about 2800 m a.s.l. for the early Pleistocene and of about 2700 m a.s.l. for the late Pleistocene (see e.g. Fan et al., 2012, for a comprehensive discussion of lake stands in the QB).

   Since the high mountain ranges are of utmost importance for the water balance of the entire QB, the question arises, how different the paleogeographic situation has been in the QB as compared with today. Although the details of tectonics are not

yet finally clarified, and vertical movements of the lithosphere have certainly influenced the QB's orography (Fang et al.,
2007) and hydrography, the results of this study are considered to be generally applicable to the mid-Pliocene, since the
paleogeographic situation has generally been similar to the presence (Dowsett et al., 2010). Since the mid-Miocene, elevations
of the QB and surrounding mountain ranges are comparable to today (Wang et al., 2008; Yuan et al., 2013). If altitudes of the
high mountain ranges in the QB would have been a few hundred meters lower than today as indicated by Fang et al. (2007),
then the negative effect of lower altitudes on annual water balance would be accompanied by the counteracting effect that
blocking of humid air masses by the high mountain ranges in the fringes of the QB would have been less strong, a fact also
studied for the entire TP (Broccoli and Manabe, 1992). Koutsodentris et al. (2019) state that their results on Late Pliocene
vegetation turnover also indicates that terrestrial ecosystem changes in Central Asia during this time period were primarily the
result of global climate change rather than of Tibetan Plateau uplift.

These statements are justified by the results obtained by applying the same methodology to the HAR data set for the 30 km
domain (see Table S9, Fig. S11 and S12 in the Supplement). Due to the coarser model grid, the altitudes of the highest
mountains are lower in the 30 km grid ($h_{max} = 5.136$ m a.s.l.) than in the 10 km grid ($h_{max} = 5.433$ m a.s.l.), and mean annual
water balance rises from -14 mm/a (10 km grid) to 3 mm/a (30 km grid). This indicates that less blocking of humid air inflow
to the QB (due to lower altitudes) overcompensates reduction in orographic precipitation in the HAR 30 km data set (due to
the coarser grid) as compared to the results for the HAR 10 km data set (higher altitudes and finer grid).

Global climate change as projected for the future and its consequences for the regional climates of China (Burke et al., 2018;
Gu et al., 2018; Hui et al., 2018) could lead to strengthening of the East Asian Summer Monsoon (Wang et al., 2008), which
could also affect the QB such that both annual specific humidity and precipitation would further increase. This would then
lead to continued recharge of groundwater reservoirs and, at a later stage, to rising lake levels or formation of new lakes, as
already observed today. In a long-term perspective, even a mega-lake system may be restored.

Assuming a slightly positive long-term mean annual water balance of 20 mm/a as discussed above, lake water levels would
rise by 100 m averaged over the entire QB within only 5000 years, which is, in a geological perspective, a very short time
period. Since water would preferentially accumulate in the low-lying areas due to surface and groundwater flows within the
QB, the effect would be even stronger in those areas, which have formerly been part of the mega-lake system. As the semi-
empirical model shows, the feedback of growing lake area on actual evapotranspiration needs to be taken into account. As
long as the lake extent is a small fraction of the entire QB, this feedback is weak. Therefore, lake growth would take place at
a high rate in the beginning. During this early stage of lake growth, the rate of mean annual lake evaporation would also be
irrelevant since lake growth would be due to runoff from land areas to the lakes.

The particular hypsometric profile of the QB is a major factor for rapid changes in lake extent as the results in Table 3 indicate.
For instance, if lake level would rise from 2672 to 2678 m a.s.l., i.e., by only 6 m, lake extent would increase from 6654 to
10.423 km$^2$, i.e., by 3769 km$^2$. Partial restoration of the mega-lake system during former Pleistocene interglacial periods as
reported by Fan et al. (2012) or Wang et al. (2012), among others, indicates that this mechanism has probably taken place
several times in Earth history.

**5 Conclusions**

This study could show that the mean annual water balance of the QB was close to zero during the 14 hydrological years from 2001 to 2014. Negative values of net precipitation prevail in the low-lying desert regions while the high mountain regions within or at the border of the drainage basin show positive values compensating water losses at lower altitudes. Orographic precipitation is strongly increasing total precipitation at higher altitudes, while actual evapotranspiration is also increasing with altitude but less strong. The latter fact, combined with high values of actual evapotranspiration over lakes, indicates that availability of water but not energy for latent heat is the main driver of actual evapotranspiration in the QB.

Annual water balance of the QB was positive during warmer years, since these years showed higher-than-normal precipitation. Annual water balance variations in the QB are driven by variations in annual precipitation while variations in annual actual evapotranspiration showed no statistically significant effects on annual water balance. The study revealed that not air temperature alone but specific humidity, in combination with air temperature, is the main climate driver of annual water balance. Inflow of moister-than-normal air to the QB takes place during warmer-than-normal years and increases both precipitation and water balance.

Even slight changes in specific humidity could induce significant changes in mean annual precipitation and subsequent changes in mean annual water balance, lake levels and lake extent. The feedback of increased total annual lake evaporation is currently weak but would be the dominant control of lake growth as soon as lakes cover a substantial fraction of the QB. This study, although not targeting on reconstruction of paleolakes or making predictions for the future, demonstrates that the maximum extent of the mega-lake system in the QB could have been sustained under climate conditions as reported by paleoclimate studies for the mid-Pliocene. The specific hypsometry of the QB makes it sensitive to rapid changes in lake extent, also under climate conditions projected for the near future.

Future research could target on acquisition of additional or improved spatially distributed data sets for water balance components and climate drivers at even higher spatial resolution to capture the details of the high mountain topography of the QB, and for longer time periods to improve assessments of environmental changes related to changes in water balance. The new global ERA5 reanalysis data set, as well as the transferability of the methods applied in this study to other regions offer new options in this respect. Other lines of research could focus on dynamical downscaling of paleoclimate simulations for Pliocene time slices or global climate projections for the future. Both kind of data sets are, in general, spatially too coarse to fully resolve atmospheric processes like orographic precipitation or actual evapotranspiration in high mountain ranges (Gu et al., 2018), such that regional water balance computations based on coarse data would, most likely, come with high uncertainties. Thus, dynamical downscaling of global atmospheric data is regarded to be essential.

Dynamical downscaling could also be used to study changes in the statistical relations as revealed in this study by artificially modified (paleo-)geographies. Since lakes tend to reduce precipitation while concurrently showing high actual evaporation, there is an upper limit for lake growth, which is, however, difficult to quantify, mainly due to remaining uncertainties of changes in precipitation that have occurred in the QB in geological history or might come with future climate change. High

spatial resolution, and thus dynamical downscaling, is also required to quantify if and how a mega-lake system would effectively recycle its own water, i.e., that atmospheric moisture stemming from lake evaporation precipitates within the basin's catchment area. Water recycling is highly important for the entire TP (Curio et al., 2015) and might also play an important, yet unknown role for the QB's water balance.


**Data availability.** The HAR data set is freely availability at http://www.klima.tu-berlin.de/HAR. The source code used in this study for analysing the HAR data is freely available upon request. The source code used for analysing the feedback of changes in lake extent and mean annual precipitation on the mean annual water balance of the QB and changes over lakes and land areas is provided in the Supplement.


**Supplement.** The supplement related to this article is available online at:

**Author contribution.** Dieter Scherer carried out the analyses and wrote the paper.

**Competing interests.** The author declares that he has no conflict of interest.

**Acknowledgments.** I would like to thank Fabien Maussion (University of Innsbruck, Austria) and Julia Curio (University of Reading, United Kingdom) for their participation in the development of the High Asia Refined analysis data set (HAR), to Erwin Appel, Svetlana Botsyun, Todd Ehlers, Sebastian Mutz (University of Tübingen), and to Marco Otto, Benjamin Schmidt, Vanessa Tolksdorf, and Xun Wang (Technische Universität Berlin, Germany) for collaboration and discussions within the Q-TiP project. My thanks also include the anonymous reviewers and the editor who substantially helped to improve the study.

**Financial support.** This work was supported by the German Research Foundation (DFG) Priority Programme 1372, "Tibetan Plateau: Formation - Climate - Ecosystems" within the DynRG-TiP project "Dynamic Response of Glaciers on the Tibetan Plateau to Climate Change" (codes SCHE 750/4-1, SCHE 750/4-2, SCHE 750/4-3), and by the German Federal Ministry of Education and Research (BMBF) Programmes "Central Asia - Monsoon Dynamics and Geo-Ecosystems" (CAME) within the WET project "Variability and Trends in Water Balance Components of Benchmark Drainage Basins on the Tibetan Plateau" (code 03G0804A), and the follow-on Programme CAME II within the Q-TiP project "Quaternary Tipping Points of Lake Systems in the Arid Zone of Central Asia" (code 03G08063C).

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

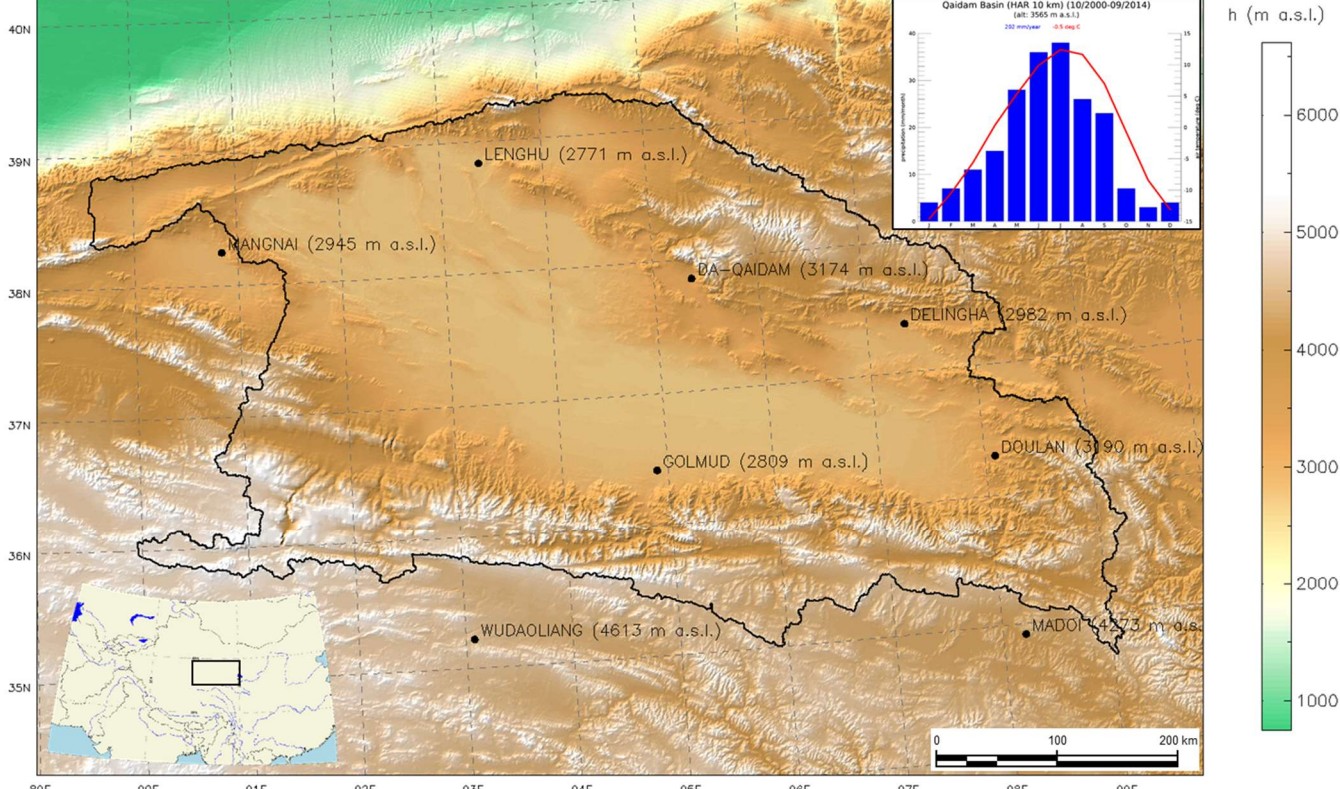

**Figure 1: Overview on the Qaidam Basin (QB), including a spatially averaged climate diagram for the QB derived from the HAR 10 km data set for the study period of 14 hydrological years (2001-2014). Black line: boundary of the QB (Lehner and Grill, 2013). Topographic shading is based on DEM data from the SRTM. Black dots indicate the locations of the eight GSOD stations within or nearby the QB.**


**Figure 2: Annual precipitation (upper left panel), actual evapotranspiration (upper right panel), and water balance (lower panel) in the Qaidam Basin (QB) during the hydrological years 2001 to 2014 derived from the HAR 10 km data set. Upper left: light grey bars: annual snowfall; dark grey bars: annual rainfall. Dotted lines: mean annual values.**


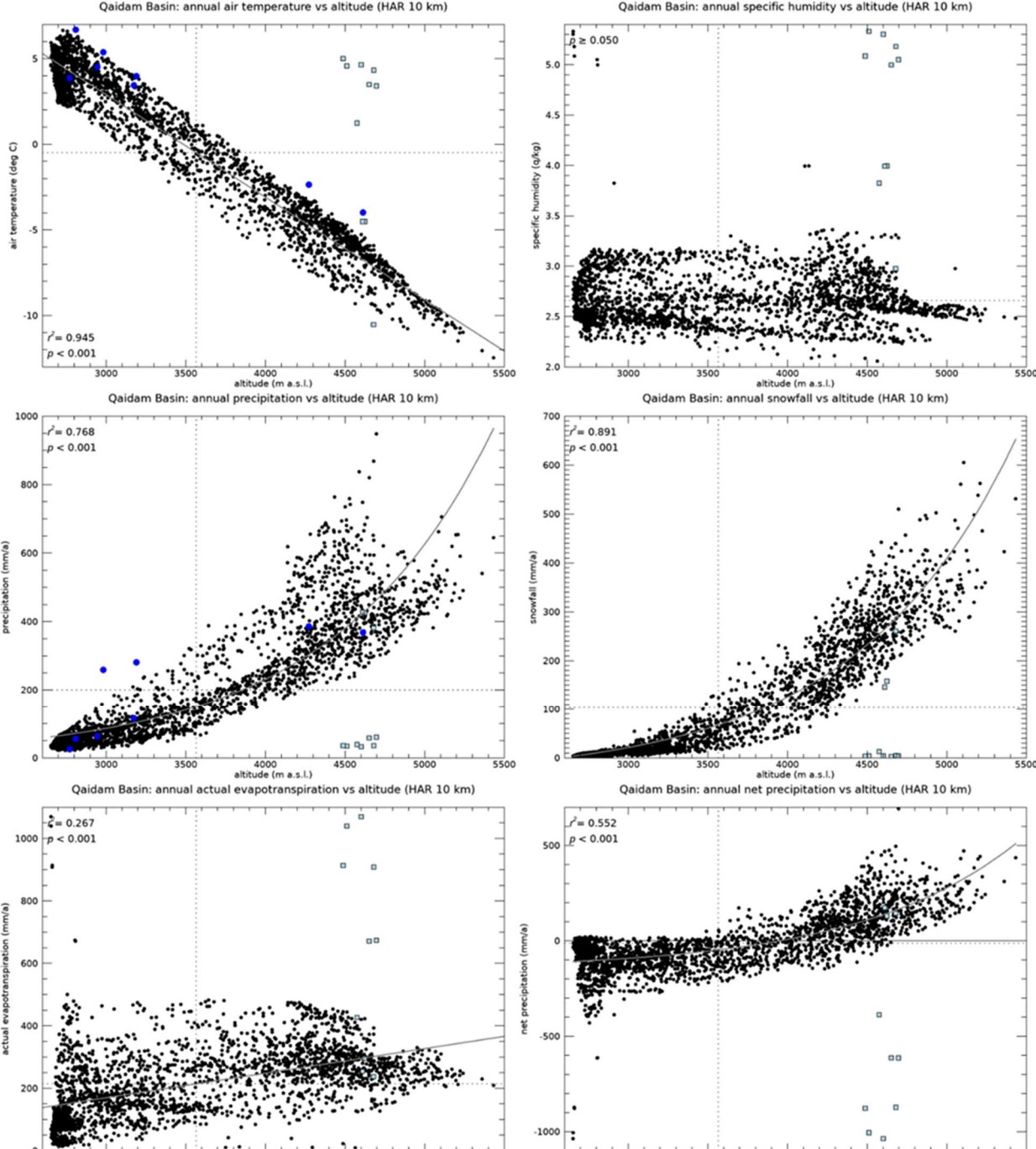

**Figure 3: Mean annual air temperature (upper left panel), specific humidity (upper right panel), precipitation (middle left panel), snowfall (middle right panel), actual evapotranspiration (lower left panel), and net precipitation (lower right panel) versus altitude in the Qaidam Basin (QB) during the hydrological years 2001 to 2014 derived from the HAR 10 km data set. Black dots: land grid points; light blue squares: grid points covered by lakes in the HAR 10 km land cover data; blue dots: data from eight GSOD stations; dotted lines: mean annual values.**


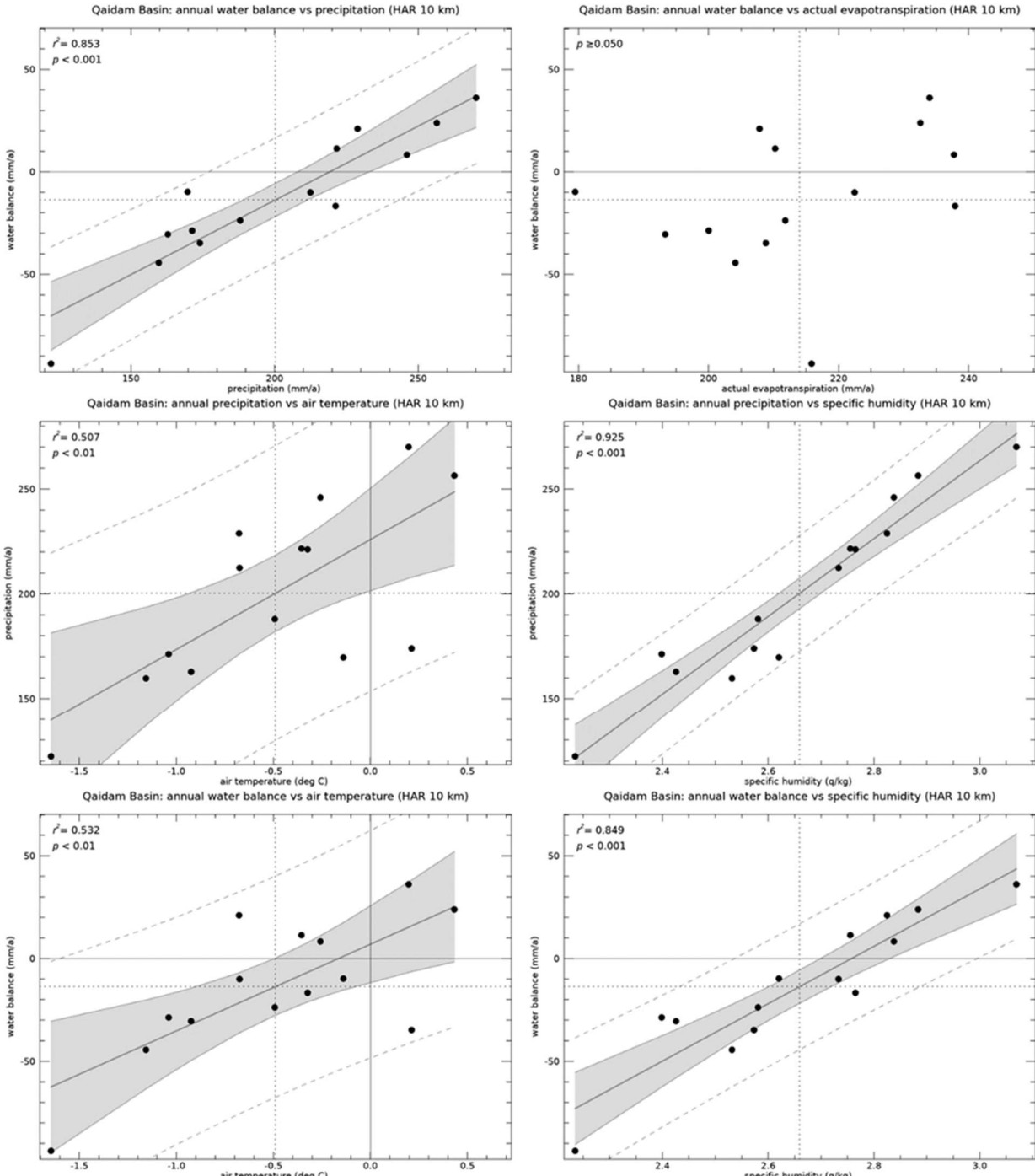

**Figure 4: Water balance versus precipitation (upper left panel) and actual evapotranspiration (upper right panel); precipitation versus air temperature (middle left panel) and specific humidity (middle right panel); water balance versus air temperature (lower left panel) and specific humidity (lower right panel) in the Qaidam Basin (QB) during the hydrological years 2001 to 2014 derived from the HAR 10 km data set. Dotted lines: mean annual values; solid lines: regression lines; light grey shades: confidence intervals; dashed lines: prediction intervals.**


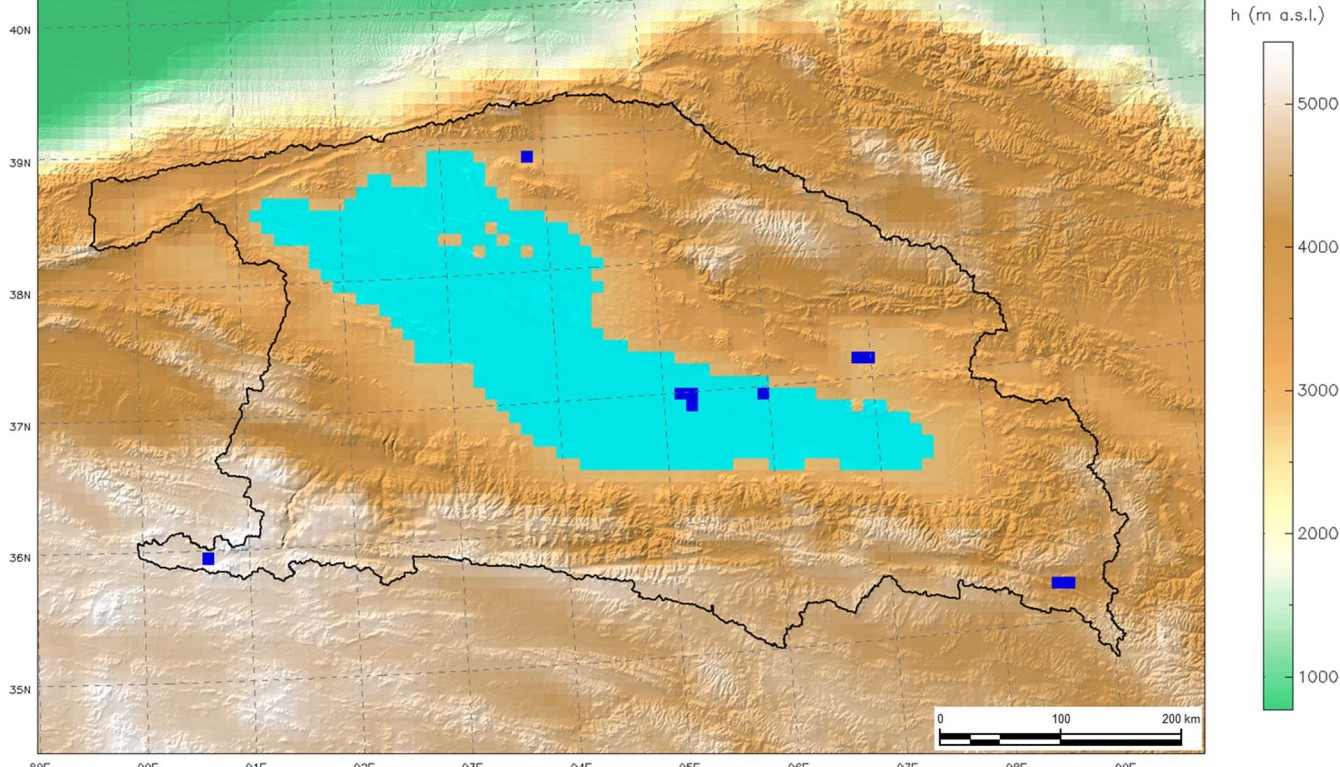

**Figure 5: Illustration of the maximum extent of the mega-lake system in the Qaidam Basin (QB) of approx. 59.000 km² as reported by Chen and Bowler (1986) using the present-day model topography of the HAR 10 km data set for accumulation of net precipitation in the QB and subsequent runoff originating from land in areas at or below 2786 m a.s.l. (marked in cyan). Black line: boundary of the QB (Lehner and Grill, 2013). Blue: present-day lake extent (1000 km²) as represented in the HAR 10 km data set. Topographic shading is based on DEM data from the SRTM.**



**Table 1: Mean monthly and annual air temperature $T$ (in deg C), specific humidity $q$ (in g/kg), precipitation $P$ (in mm/month or mm/a), rainfall $P_{rain}$ (in mm/month or mm/a), snowfall $P_{snow}$ (in mm/month or mm/a), actual evapotranspiration $ET$ (in mm/month or mm/a), and water balance $\Delta S = P - ET$ (in mm/month or mm/a) in the Qaidam Basin (QB) derived from the HAR 10 km data set; sigma: standard deviations of annual values for each quantity during the hydrological years 2001 to 2014.**

| month | 10 | 11 | 12 | 1 | 2 | 3 | 4 | 5 | 6 | 7 | 8 | 9 | **year** | sigma |
|---|---|---|---|---|---|---|---|---|---|---|---|---|---|---|
| $T$ | -0.7 | -8.4 | -13.1 | -14.6 | -10.5 | -5.6 | 0.3 | 5.3 | 9.9 | 12.4 | 11.6 | 7.0 | **-0.5** | 0.6 |
| $q$ | 2.2 | 1.4 | 1.0 | 0.9 | 1.2 | 1.5 | 1.9 | 2.8 | 4.3 | 5.6 | 5.1 | 3.9 | **2.7** | 0.2 |
| $P$ | 7 | 3 | 4 | 4 | 7 | 11 | 15 | 28 | 36 | 38 | 26 | 23 | **200** | 43 |
| $P_{rain}$ | 1 | 0 | 0 | 0 | 1 | 1 | 4 | 9 | 21 | 30 | 19 | 12 | **97** | 31 |
| $P_{snow}$ | 6 | 3 | 4 | 4 | 6 | 10 | 11 | 19 | 15 | 8 | 7 | 11 | **103** | 17 |
| $ET$ | 13 | 8 | 5 | 5 | 9 | 15 | 18 | 25 | 29 | 34 | 31 | 21 | **214** | 18 |
| $\Delta S$ | -6 | -4 | -1 | -2 | -2 | -4 | -4 | 2 | 7 | 4 | -5 | 1 | **-14** | 34 |


**Table 2: Estimated changes in annual air temperature $\Delta T$ (in deg C), specific humidity $\Delta q$ (in g/kg), precipitation $\Delta P$ (in mm/a), and water balance $\Delta(\Delta S)$ (in mm/a) in the Qaidam Basin (QB) with respect to present conditions, and resulting estimated annual water balance $\Delta S$ (in mm/a) for the mid-Pliocene. Red colours: estimates based on changes $\Delta T$; blue colours: estimates based on changes $\Delta q$; black colours: estimates based on changes $\Delta P$. Input values for $\Delta T$ (magenta shades) and $\Delta P$ (grey shades) are estimates based**
**on the studies of Zhang et al. (2013c) and Mutz et al. (2018), while present values and sensitivities are derived from the HAR 10 km data set during the hydrological years 2001 to 2014 as discussed in the text.**

| sensitivity | value | unit | $\Delta T$ min | max | $\Delta q$ min | max | $\Delta P$ min | max | $\Delta(\Delta S)$ min | max | $\Delta S$ min | max | mean |
|---|---|---|---|---|---|---|---|---|---|---|---|---|---|---|
| $\dfrac{\partial q}{\partial T}$ | 0.288 | g kg$^{-1}$ K$^{-1}$ | **1.0** | **2.0** | 0.3 | 0.6 | | | | | | | |
| $\dfrac{\partial P}{\partial T}$ | 52.40 | mm a$^{-1}$ K$^{-1}$ | | | | | 52 | 105 | | | | | |
| $\dfrac{\partial \Delta S}{\partial T}$ | 42.17 | mm a$^{-1}$ K$^{-1}$ | | | | | | | 42 | 84 | 28 | 70 | **49** |
| $\dfrac{\partial P}{\partial q}$ | 185.6 | mm a$^{-1}$ g kg$^{-1}$ | | | 0.3 | 0.6 | 53 | 107 | | | | | |
| $\dfrac{\partial \Delta S}{\partial q}$ | 139.7 | mm a$^{-1}$ g kg$^{-1}$ | | | | | | | 40 | 81 | 26 | 67 | **46** |
| $\dfrac{\partial \Delta S}{\partial P}$ | 0.725 | mm a$^{-1}$ mm$^{-1}$ a$^{-1}$ | | | | | **50** | **100** | 36 | 73 | 22 | 59 | **40** |

**Table 3: Mean annual rate of lake evaporation $ET_{lake}$, change in mean annual precipitation $dP_{QB}$ with respect to present-day precipitation (HAR 10 km; 200 mm/a), lake extent $A_{lake}$, and approximate altitude of lake level $z_{lake}$ for equilibrium lake states in the Qaidam Basin (QB) as computed by a semi-empirical model (cf. R source code in the Supplement). Values in column "HAR" are taken from the HAR 10 km data set and represent mean annual values for the hydrological years from 2001 to 2014. Table 3 also displays a combination of three projections for $ET_{lake}$ (600, 800 and 1000 mm/a) and two projections for $dP_{QB}$ (50 and 100 mm/a). For each of the three projections for $ET_{lake}$, the value for $dP_{QB}$ is shown that would sustain the maximum extent of the mega-lake system in the QB (see Fig. 5). Values for $z_{lake}$ have been computed from the HAR 10 km model topography by choosing the value that produces a lake extent closest to the equilibrium lake extent as computed by the semi-empirical model.**

| quantity | unit | HAR | projections | | | | | | | | |
|---|---|---|---|---|---|---|---|---|---|---|---|
| $ET_{lake}$ | (mm/a) | 649 | 600 | | | 800 | | | 1000 | | |
| $dP_{QB}$ | (mm/a) | 19 | 50 | 100 | 210 | 50 | 100 | 270 | 50 | 100 | 330 |
| $A_{lake}$ | km$^2$ | 1000 | 10423 | 25773 | 59000 | 8067 | 19580 | 59000 | 6654 | 15864 | 59000 |
| $z_{lake}$ | m a.s.l. | 2653 | 2678 | 2711 | 2786 | 2674 | 2698 | 2786 | 2672 | 2688 | 2786 |