# Peer review of "Survival of the Qaidam Mega-Lake System under Mid-Pliocene Climates and its Restoration under Future Climates"

_Hydrology and Earth System Sciences, 2019_

## Referee Comment (RC1) · Anonymous Referee #1 · 5 Nov 2019

This paper assessed the water balance in Quaidam basin, where the mega-lake existed in mid-Pliocene, using High Asia Refined analysis data during 2001-2014. The results showed almost zero annual balance with positive during warmer and negative during dry years. Also the altitudinal tendencies of climate parameters with their contribution to the water balances are diagnosed by simple regression (scattering) analysis. Assessments of annual water balance in the semi-dry and endorheic basin behind TP, using comprehensive data based on satellite estimates and numerical model, are challenging. If the trend shown in Fig.2 could be verified by independent data or evidences in the social activities, the budget assessments would be reliable and useful in the present climate condition. Also, in the scattering diagram in Fig. 4, years with

far from linier regression should be diagnosed intensively as Fig. 3 to know the factors in HAR data that caused positive or negative budget (e.g. as mentioned in P6L175). Besides, there are many fundamental unintelligible and not logical parts as following comments.

1) Mountain water (including from glaciers) are accumulated in the underground, and foster the society and ecosystem (used by human or biosphere/agriculture) in the semi-arid basins by pumping up especially during non-rainy season. This part is ignored in (1) and flowing analysis. Lake is the ground water level over the surface, but it is very strange that author neglected the groundwater matter (P2L34) and discusses about the lake existence in the past. 2) There is a huge time scale gap between the 10 years time slice for present climate (2001-2014) and a time slice of Midïij■Pliocene (3.3-3.0 Ma=30,000,000 years). Author also recognized this issue in P8L234. I can not understand the logic of such comparison. It is very nonsense to compare 10 years/annual average to the paleo climate time scale. If your focus is the mechanism of mega-lake formation and maintenance for several millions of years, water budget simulation during the mid-Pliocene is necessary with boundary condition that fits proxy data. Also, the evaporation over the lake water is quite different from the desert surface that is not considered in your study. Dry soil/sand at the skin surface blocks soil moisture movement from the underground. 3) I could not understand that what and how the author estimates in P7L193. I speculated that statistical relations between altitude and meteorological parameters derived in Fig. 3 was performed as functions of future or passed expected temperature differences. If so, the methods are wrong. Relations in Fig. 3 was derived from the dynamical downscaling in the present climate condition, and does not work in the passed or future climate condition without simulating the similar dynamical downscaling under Mid-Pilocene global climate condition (see PlioMIP2 project etc.). 4) Paper structure is very strange. Results (figures) are only limited in the statistical relations and data aggregation using the 10 years HAR data, without clear figures to explain that why the mega-lake could form/sustain during the mid-Pliocene. Conclusion do not contain the main results but his own theory (idea) was extended,

and abstract mentioned that analogue of Mars could fit to the study results without any analysis in the main contents.

Minor comments P2L42ãĂĂWhich is the "high mountain range in the Qaidam basin"? Is this for Midïij■Pliocene? P2L47-50 Is this objective parts? Can not understand the discussion. P3L74 How many GSOD stations in the target area ? Quite few? Or many? Black point in Fig. 1? P3L85 Fig. 3 comes before Fig. 2 ? P4L91 climate driver -> variables/elements ? P6L164-175 Is this review ? Then better to move in Chapter 1. P7L199ãĂĂ"comparative with„" L205ãĂĂ"almost identical", very vague terms and I can not understand. P7L210 Some papers show that climate in Mid-Pilocene is warm and wet (e.g. by Zhang). Celements et al. (1996) shows that nonstationary phase of Asian monsoon during Plio-Pleistoce, so is it sure that mega-lake was stable for several Ma years? P7L220 "blocking humidity" Thermal effect of TP causes subsidence around the northeast Asian area to form dry climate including around Qaidam basin. See Sato and Kimura, 2005, GRL for instance. Uplift of northwestern Tibet in the target era may also effect to this effect and also changes intensity/route of westerly disturbances.

---

## Referee Comment (RC2) · Anonymous Referee #2 · 8 Nov 2019

The author carried out a study about the survival of the Qaidam mega-lake system in Pliocene by analyzing the modern water balance of the basin. The author finds that the water balance of Qaidam basin is nearly zero under present climate condition and believes that Qaidam basin may switch from negative to positive in the near future. Although the story is quite interesting, there is a lack of robust evidence. In particular, the time scale of data used in the analysis is so different from a geological epoch, and therefore their connection is reluctant. I do not know how robust this linear speculation between the current and past lake conditions could be. As far as my knowledge, there are several fundamental flaws.

[Figure]

1. Line 32, lake evaporation is missing in the equation. Lake evaporation is very different from land evaporation. Although it is very small in modern time because the lake area is small (only ~1000 km2), it can be much large when the lake area is hundred times (~120, 000 km2) in Pliocene. Moreover, it is hard to derive the water balance in such a case from current HAR data, considering very different effects of a large lake area that is not included in HAR data. Lake evaporation is neither considered in the remaining discussion. For example, in line 227-231, the author did not consider the impact of lake evaporation. In fact, when the lake continues to increase, lake evaporation increases and an equilibrium between input and output will be reached. Therefore, lake water level would not rise by 400 m over the basin within only 10 ka, even there is a positive long-term mean annual water balance.

2. Line 166-174, as the author said, both lake area and lake number in the Qaidam basin have increased from the last two decades. The groundwater in the Qaidam basin was recharged between 2003 and 2012 due to changes in terrestrial water storage of 20.6 km3 at rate of 8 mm/a. These studies indicate that there should be considerable positive water balance during the last two decades in Qaidam basin, which contrasts with the main conclusion of the study, which shows the water balance of the basin is almost zero (Line 174).

3. L194-197: "the estimated changes in precipitation due to changes in air temperature would be 52 to 105 mm/a...The mean change in water balance as inferred from the changes in air temperature would lead to a positive mean annual water balance of 49 mm/a." I cannot believe this derivation. The positive water balance is too big, because most of precipitation would be lost through evaporation in such a dry environment.

4. The main conclusion of the study is 'near-future climates not much different from present conditions could cause rising lake levels and expanding lake areas, and may result in restoration of the Qaidam mega-lake system over geological time scales' (line12-14). Although I am not a paleo-climatologist, I guess the formation of mega-lake system over geological time scales might have underwent special climate conditions (e.g. different scales of lake-air interaction, different land cover condition), and this speculation is quite uncertain. Even current climate yields positive mass balance, this positive balance would be lost soon due to the increase of lake evaporation when lake expands.
* * *

---

## Editor Comment (EC1) · Bob Su (Editor) · 9 Jan 2020

(Review comments by referee #3, uploaded by the editor)

Review Survival of the Qaidam Mega-Lake system under Mid-Pliocene Climates and its restoration under Future Climates and its restoration under Future climates. By: Dieter Scherer

The paper provides a nice straightforward estimation of the regional water balance of the Qaidam basin. It is interesting to observe that the water balance calculations confirm that the balance is near zero. It implies that although parts of the basin has

hyper arid conditions, there is apparently a zero balance. This as such answers the question: can a lake exist for prolonged periods in this basin under arid conditions. I am struggling with the way this is framed. We have no future time series so we can only speculate what will happen in the future. It is interesting that the current trends of climate change seems to lead to a positive water balance. The problem I have is that this does not automatically imply that the lake level will rise. I think the claim of a tipping point (line 15) is not substantiated, it could be a threshold but a tipping point suggest a complete new system equilibrium. This can not be predicted with linear regressions based on the current system dynamics. The link with the Mega Lake is also unclear. No information of its extent, depth etc is given. The existence of such a lake would invalidate many of the assumptions now made to calculate the water balance. Please discuss possible feedbacks in the studied system. You now seem to assume they dot not exist, which I consider unlikely.

Minor comments: Line 7: the phrase increasingly arid climates is confusing given the fact that the analysis is not suitable to deal with large long term climate changes Line 14: the restoration of the mage-lake is very speculative given the fact that the calculations are based on 14 years of data. Such a restoration would need ten thousands of years. Line 24: the disappearance of the lake during the last 100 ka is intriguing. What was different? Line 30-40: please provide units and support (10*10 km for example) Line 45-50: please formulate a concrete testable hypothesis. I propose that within the Qiadam basin the water balance is near zero. Line 74: what are valid data? How is this supported by evidence? Line 82-84: is there possibly spatial autocorrelation in your analysis? Line 103: are you implying that there are feedbacks in the system? If this is the case, it implies that you should be very carful to use your regression relationship outside current conditions. Line 120: an important conclusion. Please emphasize this more in abstract and conclusions Line 134: same as above important insight in current system Line 174: how sensitive is your study for spatial resolution? Line 197: how long would it take to fill up the whole mage-lake (with this rate of 49 mm/a) Line 238: how important can the orographic precipitation be? You now assume this not to happen. Ad

how does the presence of a big lake affect this effect?

---

## Author Response (AR2)

**General remarks**

Thanks to the valuable comments of the three reviewers and the editor, I have revised the manuscript as explained below. I have also followed the requests of the editorial office to check and correct typos, phrasing, figure and table captions, formulas, as well as texts and legends in figures and tables.

**Replies to Reviewer 1**

This paper assessed the water balance in Quaidam basin, where the mega-lake existed in mid-Pliocene, using High Asia Refined analysis data during 2001-2014. The results showed almost zero annual balance with positive during warmer and negative during dry years. Also the altitudinal tendencies of climate parameters with their contribution to the water balances are diagnosed by simple regression (scattering) analysis.

Reply: The study addresses both the state of the Qaidam basin's water balance during present-day conditions, and its sensitivity to variations in mean annual near-surface air temperature and humidity spatially averaged over the basin. These physical quantities are addressed in regional paleoclimate studies where uncertainties in global-scale forcing of regional climates and in paleogeographic data do not allow for spatially detailed (high-resolution) analyses of climate and hydrology. This holds particularly true for orographically induced precipitation which is not resolved by coarse grids used in paleo-climate modelling but is of utmost importance as my study reveals.

Changes: Text improved.

Assessments of annual water balance in the semi-dry and endorheric basin behind TP, using comprehensive data based on satellite estimates and numerical model, are challenging. If the trend shown in Fig.2 could be verified by independent data or evidences in the social activities, the budget assessments would be reliable and useful in the present climate condition.

Reply: I did not analyse trends. Moreover, I examined how variations in the basin's water-balance relate to its components, i.e., precipitation and actual evaporation, and to variations in near-surface air temperature and specific humidity. All physical quantities are resolved by the High Asia Refined analysis data set (HAR) on a strict physical basis using observational data that have been thoroughly examined and widely used in a multitude of independent scientific studies. All data including the HAR data set are freely available to the public, such that reproducibility of the results of this study is ensured. The discussion part comprises various studies which are compared with my own results. I will add more references to scientific articles showing that the results of this study are not in contradiction to other, independent studies.

Changes: Text improved; further references added.

Also, in the scattering diagram in Fig. 4, years with far from linier regression should be diagnosed intensively as Fig. 3 to know the factors in HAR data that caused positive or negative budget (e.g. as mentioned in P6L175).

Reply: I don't understand what is requested by the reviewer. The sensitivity of a dependent variable $y$ on variations of one of the independent variables $x$ is given by the partial derivative $\partial y/\partial x_i$ where $x_i$ denotes the independent variable to be analysed while all the other independent variables are held constant. This is exactly what simple linear regressions provide as result, since the regression coefficient is the partial derivate. Statistical significance testing is a common method to ensure that the probability that a correlation could be purely by chance is below a certain threshold (here 0.05). I only showed sensitivities that are statistically significant. Of course, correlation is not causation. But the physics in the WRF model allows for in-depth analyses of the reasons behind correlations.

Changes: Text improved.

Besides, there are many fundamental unintelligible and not logical parts as following comments.

Reply: This statement indicates that a revised version of the manuscript should better explain the concepts and methods behind the study, and that additional studies should be added to the text.

Changes: Text improved; further references added.

1) Mountain water (including from glaciers) are accumulated in the underground, and foster the society and ecosystem (used by human or biosphere/agriculture) in the semi-arid basins by pumping up especially during non-rainy season. This part is ignored in (1) and flowing analysis.

Reply: I have explained in the manuscript that the study addresses water balance in total but not the individual storage systems separately. The latter would be beyond the scope of my study, and would not be feasible to analyse just by any atmospheric data set. As shown in Eq. (1), the water balance of a large endorheic basin can be approximated by net precipitation, i.e., the difference between precipitation and actual evapotranspiration, since lateral fluxes of surface water and groundwater at the basin's borders are excluded by definition ($\Delta Q_{sfc} = 0$), or are negligible when compared to net precipitation integrated over the entire basin. Thanks to the reviewer's comment, I detected an error in Eq. (1) that needs to be corrected in a revised version of the manuscript: the term $\Delta Q_{sub}$ (indicating subsurface) in the equation must be replaced by $\Delta Q_{gw}$, which is the term used in the text for changes in groundwater storage.

In addition, I will change the terminology in the revised manuscript, such that the term water balance is only used when referring to the spatial average of net precipitation over the entire basin. Net precipitation shall be used when referring to individual grid points or areas within the basin, because then, lateral fluxes inside the basin are correctly treated.

Many studies use the term terrestrial water storage (TWS), which also does not differentiate between different hydrological subsystems. As long as water that is exchanged between subsystems (glaciers, lakes, groundwater, etc.) resides inside the basin it does not change TWS. Thus, changes in TWS of the Qaidam basin are directly caused by the spatial average of net precipitation, i.e., water balance. I will add further studies and revise the text to make the concept better understandable.

Changes: Text improved; terminology and formula adopted; further references added.

Lake is the ground water level over the surface, but it is very strange that author neglected the groundwater matter (P2L34) and discusses about the lake existence in the past.

Reply: I do, of course, know the definition of a lake. I have discussed groundwater in general, and changes in groundwater storage in particular, in the manuscript, e.g. in Eq. (1); see above. In the discussion part (P6 L171-174), I discussed changes in groundwater reservoirs, and I referred to a study from Jiao et al. (2015) that showed the accordance of my results with those from an analysis of GRACE satellite data. I will give more details on this study and add further studies like the study of Loomis et al. (2019) to the discussion part:

Loomis BD, Richey AS, Arendt AA, Appana R, Deweese Y-JC, Forman BA, Kumar SV, Sabaka TJ, and Shean DE, 2019. Water Storage Trends in High Mountain Asia. *Front. Earth Sci.* 7, 235. doi: 10.3389/feart.2019.00235.

Over long time periods, positive changes in TWS will result in rising groundwater levels, and subsequently, in rising lake levels. Both phenomena are currently taking place in the Qaidam basin as reported in the literature referenced in the manuscript. My study does, however, not address the timing of storage changes in the different reservoirs.

Changes: Text improved; further references added.

2) There is a huge time scale gap between the 10 years time slice for present climate (2001-2014) and a time slice of Midïij Pliocene (3.3-3.0 Ma=30,000,000 years). Author also recognized this issue in P8L234. I can not understand the logic of such comparison. It is very nonsense to compare 10 years/annual average to the paleo climate time scale.

Reply: The study does not intend to reconstruct the water balance of the Qaidam basin during the mid-Pliocene epoch. This is indeed impossible since no suitable paleogeographic or paleoclimate data required for detailed hydrological analyses or modelling studies are available, today. Moreover, I analysed the sensitivity of the present-day water balance (14 hydrological years) of the entire Qaidam basin to variations in spatially averaged mean annual near-surface air temperature and specific humidity. Then, I used results from independent paleoclimate studies on the Qaidam basin for the mid-Pliocene epoch (both from proxy-based and numerical modelling studies) to illustrate that the long-term mean water balance would have been positive during this period. The rationale behind this approach is given by the fact that physical laws do not change over time, and the results are in accordance to independent findings by the proxy-based studies that show the existence of a mega-lake system during a period of intensifying aridification of the region. I will add further details and additional references on this topic. I also discussed the assumptions and limitations of this approach (P7 L210-221).

Changes: Text improved; further references added.

If your focus is the mechanism of mega-lake formation and maintenance for several millions of years, water budget simulation during the mid-Pliocene is necessary with boundary condition that fits proxy data.

Reply: As stated above, this study is not targeting on reconstruction of the water balance, and, even more, not on modelling the hydrology of the Qaidam basin during the mid-Pliocene epoch.

Changes: Text improved.

Also, the evaporation over the lake water is quite different from the desert surface that is not considered in your study. Dry soil/sand at the skin surface blocks soil moisture movement from the underground.

Reply: Actual evapotranspiration of lake water is included in the study. Spatial averages of all physical quantities also comprise grid points indicated as lakes in the HAR data set. I also discussed the specific role of lakes in P4 L116-119. In addition, I stated in P8 L242-246 that there are processes related to evaporation of lake water and its consequences for water balance and lake-level changes not yet solved, and thus require further studies. Evaporation of the desert surfaces are also included in the study. Soil moisture is one of the input variables provided by the global operational FNL data set. Even if soil properties would not be accurately represented in the WRF model, the very low soil moisture values nevertheless limit evaporation. If aridity resistances would be larger as indicated by the reviewer, then net precipitation would be even higher (less negative or more positive), and thus supporting the findings of this study.

Changes: Text improved.

3) I could not understand that what and how the author estimates in P7L193. I speculated that statistical relations between altitude and meteorological parameters derived in Fig. 3 was performed as functions of future or passed expected temperature differences. If so, the methods are wrong.

Reply: The values for minimum and maximum changes in long-term mean annual air temperature (1 and 2 K) are taken from the literature as referenced in the manuscript. These values are relative to recent mean annual air temperature, and are conservative estimates for both the mid-Pliocene and for future climate conditions. They are not reconstructions or projections in a strict sense, moreover they show the range of possible changes in mean annual air temperature that are generally compatible with the results from various independent studies. I will revise the text to make the approach better understandable, and I will add further studies.

Altitude is not explicitly considered in this part of the study. All values are spatial averages for the respective quantities, thus representing the whole spectrum of altitudes in the basin. Also, the sensitivities are derived from the simple regressions for spatial averages of mean annual values of the respective quantities.

Changes: Text improved; terminology changed; further references added.

Relations in Fig. 3 was derived from the dynamical downscaling in the present climate condition, and does not work in the passed or future climate condition without simulating the similar dynamical downscaling under Mid-Pilocene global climate condition (see PlioMIP2 project etc.).

Reply: As stated above, the study does not intend to reconstruct past and future climates and water balance.

Changes: Text improved; further references added.

4) Paper structure is very strange. Results (figures) are only limited in the statistical relations and data aggregation using the 10 years HAR data, without clear figures to explain that why the mega-lake could form/sustain during the mid-Pliocene.

Reply: As stated above, the study does not intend to reconstruct past and future climates and water balance. The main part of the study is to analyse 14 years of spatially and temporally detailed data from the HAR10 to quantify water balance of the entire basin, the importance of the high-altitude parts for the water balance of the entire basin, and the sensitivity of the water balance of the entire basin to variations in mean annual air temperature and specific humidity spatially averaged over the entire basin as presented in the results part. Results are then discussed with respect to uncertainties, and compared to results from other independent studies.

If requested by the reviewer, I could move the results of the sensitivity study to the results part. However, I put this into the discussion part in order to first discuss that the results of the HAR-based analyses are consistent with the findings from independent studies and thus applicable, before using them for assessing potential changes of the water balance in the past and the future.

Changes: Text improved; subsections added to Results and Discussion sections to improve the structure and readability of the manuscript.

Conclusion do not contain the main results but his own theory (idea) was extended, and abstract mentioned that analogue of Mars could fit to the study results without any analysis in the main contents.

Reply: I will repeat the main results in the conclusions part in more detail in a revised version of the manuscript. The reference to the Qaidam basin discussed as analogue to Mars is just illustrating the discrepancy between the hyper-aridity of the low-altitude parts of the basin and the water balance of the entire basin, which is an integral over the low- and high-altitude parts. There are many studies that focus just on the low-altitude parts disregarding the importance of the high-mountain regions for the water balance.

Changes: Paragraph summarizing the main findings of this study added to the Conclusions section; text in Conclusions section improved; text parts shifted to the Discussion section.

Minor comments P2L42ã ˘A ˘AWhich is the "high mountain range in the Qaidam basin"? Is this for Midïij Pliocene?

Reply: As shown in Fig. 3 and discussed in text (P4 L113-115), net precipitation is positive, on average, at altitudes above 4000 m asl. The term water balance will be avoided in a revised version of the manuscript since it is misleading as discussed above. The sensitivity computations do not consider changes in altitudes, but the discussion part contains a paragraph (P7 L210-221) that refers to Fang et al. (2007) who report on paleo-altitudes of the Qaidam basin slightly lower than today. In addition, I have included results based on HAR30 data, in which altitudes are lower than in the HAR10 data due to the coarser grid. These results show that lower altitudes lead to less blocking of atmospheric water transport, and thus, to slightly higher water balance values. I will add further studies on this topic in a revised version of the manuscript.

Changes: Text improved; terminology changed; further references added; HAR 30 km results better explained.

P2L47-50 Is this objective parts? Can not understand the discussion.

Reply: The last sentence of this paragraph (P2 L46-49) is one of the main motivations behind this study. The literature referenced in this sentence essentially says that environmental changes that could eventually take place in the future due to global climate change could be also studied by investigations that address the mid-Pliocene epoch. The proxy-based studies showed that a mega-lake was present during this time, thus the objective of the study was to identify a physically based mechanism that could explain the existence of the mega-lake in the past, and could also assist to estimate potential future environmental changes.

Changes: Text improved.

P3L74 How many GSOD stations in the target area ? Quite few? Or many?

Reply: As written in the text, there are eight GSOD stations within or nearby the basin. None of them is located at very high altitudes, i.e., the altitude of the highest GSOD station (Wudaoliang; see Fig. 1) is just 4613 m asl. Data scarcity was one of the major reasons for developing the HAR data set as explained by Maussion et al. (2011, 2014).

Changes: Text improved.

Reply: Thank you! The black points are the eight GSOD stations (with names and altitudes) within or nearby the Qaidam basin. I will add this information to the figure caption.

Changes: Figure caption improved.

Reply: Thank you! I overlooked that I mentioned Fig. 3 before Fig. 2 in the data and methods part. I will change the sequence of the figures accordingly.

Changes: Reference to Fig. 3 removed in the Data and methods section.

Reply: In the text, climate drivers are air temperature and humidity. I will revise the text such that the term climate driver is avoided.

Changes: Term climate driver is kept but better explained.

Reply: This paragraph belongs to the discussion, because the results of this study are compared to findings from other studies.

Changes: Text improved.

P7L199ã ˘A ˘A"comparative with,," L205ã ˘A ˘A"almost identical", very vague terms and I cannot understand.

Reply: This text shall be revised to better explain the results shown in Table 2.

205  Changes: Text improved.

P7L210 Some papers show that climate in Mid-Pilocene is warm and wet (e.g. by Zhang).

Reply: Yes, and I have referenced the studies in P6 L175-188.

Changes: None.

210

Celements et al. (1996) shows that nonstationary phase of Asian monsoon during Plio-Pleistoce, so is it sure that mega-lake was stable for several Ma years?

Reply: I only refer to the mid-Pliocene epoch but not to later stages, where the mega-lake system started to shrink, as described in the introduction part (P1 L19-24). I will add further studies to the introduction.

215  Changes: Text improved; further references added.

P7L220 "blocking humidity" Thermal effect of TP causes subsidence around the northeast Asian area to form dry climate including around Qaidam basin. See Sato and Kimura, 2005, GRL for instance. Uplift of northwestern Tibet in the target era may also effect to this effect and also changes intensity/route of westerly disturbances.

220  Reply: As noted above, I will add further studies on this topic in a revised version of the manuscript, and I will better explain the findings from the HAR30 analysis.

Changes: Text improved; further references added; HAR 30 km results better explained.

**Replies to Reviewer 2**

225  The author carried out a study about the survival of the Qaidam mega-lake system in Pliocene by analyzing the modern water balance of the basin. The author finds that the water balance of Qaidam basin is nearly zero under present climate condition and believes that Qaidam basin may switch from negative to positive in the near future.

Reply: The study addresses both the state of the Qaidam basin's water balance during present-day conditions, and its sensitivity to variations in mean annual near-surface air temperature and humidity spatially averaged over the basin. I as the author do

230  not believe anything; I am just reporting and interpreting the results of the study, which is based on data that has been rigorously validated against independent observations. In particular, I do not speculate on the evolution of the Qaidam basin under future climates, moreover I showed that the basin's water balance is highly sensitive to changes in mean annual air temperature and specific humidity such that it is physically possible that the mega-lake system could be restored on geological time scales, as it has happened several times in the past as independent proxy-based studies have revealed.

235  Changes: Text improved.

Although the story is quite interesting, there is a lack of robust evidence. In particular, the time scale of data used in the analysis is so different from a geological epoch, and therefore their connection is reluctant.

Reply: The study does not intend to reconstruct the water balance of the Qaidam basin during the mid-Pliocene epoch. This is indeed impossible since no suitable paleogeographic or paleoclimate data required for detailed hydrological analyses or modelling studies are available, today. Moreover, I analysed the sensitivity of the present-day water balance of the entire Qaidam basin to variations in spatially averaged mean annual near-surface air temperature and specific humidity. Then, I used results from independent paleoclimate studies on the Qaidam basin for the mid-Pliocene epoch (both from proxy-based and numerical modelling studies) to illustrate that the long-term mean water balance would have been positive during this period. The rationale behind this approach is given by the fact that physical laws do not change over time, and the results are in accordance to independent findings by the proxy-based studies that show the existence of a mega-lake system during a period of intensifying aridification of the region. I will add further details and additional references on this topic. I also discussed the assumptions and limitations of this approach (P7 L210-221).

Changes: Text improved; further references added.

I do not know how robust this linear speculation between the current and past lake conditions could be. As far as my knowledge, there are several fundamental flaws.

Reply: I don't speculate but carried out a study that strictly follows the principles of transparency and reproducibility. If the reviewer questions the correctness of my analyses he/she might download the freely available data and check my statistical analyses. The reviewer might also give reference to studies that contradict my findings. Again, it is important to recognise that the study does not try to reconstruct the water balance of the mid-Pliocene epoch. In this case, I would, indeed, have to prove that the current sensitivity would have been the same in the past. I only showed with my study, that the current sensitivity, which is statistically significant and in accordance with findings from independent studies, would be able to explain how the mega-lake system has survived under the still very dry climates of the mid-Pliocene epoch. I also discussed the limitations of this approach, but I realise from the comments of the reviewers that the whole concept behind the study needs to be better explained, and that the results need to be further substantiated by adding more references.

Changes: Text improved; further references added.

1. Line 32, lake evaporation is missing in the equation. Lake evaporation is very different from land evaporation. Although it is very small in modern time because the lake area is small (only ~1000 km2), it can be much large when the lake area is hundred times (~120, 000 km2) in Pliocene. Moreover, it is hard to derive the water balance in such a case from current HAR data, considering very different effects of a large lake area that is not included in HAR data. Lake evaporation is neither considered in the remaining discussion. For example, in line 227-231, the author did not consider the impact of lake evaporation.

270    Reply: Evaporation of lake water is included in the study. Spatial averages of all physical quantities also comprise grid points indicated as lakes in the HAR data set. I also discussed the specific role of lakes in P4 L116-125. In addition, I stated in P8 L242-243 "…*Since lakes tend to reduce precipitation while concurrently showing high actual evaporation, there should be an upper limit for lake growth, which is, however, not yet known*…" If the reviewer is aware of any study that quantifies lake evaporation of the mega-lake system of the Qaidam basin (or a mega-lake in any other basin) during the Pliocene, I would be

275    very happy to get knowledge about the results. There must definitely be such a negative feedback mechanism limiting lake growth, and it would be of utmost importance for reconstructing the evolution the mega-lake system to know more about the details. I have also mentioned in P8 L243-246 that regional water recycling could counteract this feedback, but the quantitative role of this process is also still unknown. Thus, I concluded that further studies are required on these topics.

   As shown in Eq. (1), the water balance of a large endorheic basin can be approximated by net precipitation, i.e., the difference

280    between precipitation and actual evapotranspiration, which also comprises lake evaporation. Thanks to the comments of reviewer 1, I detected an error in Eq. (1) that needs to be corrected in a revised version of the manuscript: the term $\Delta Q_{sub}$ (indicating subsurface) in the equation must be replaced by $\Delta Q_{gw}$, which is the term used in the text for changes in groundwater storage. I will also change the terminology in the revised manuscript, such that the term water balance is only used when referring to the spatial average of net precipitation over the entire basin. Net precipitation shall be used when referring to

285    individual grid points or areas within the basin, because then, lateral fluxes inside the basin are correctly treated.

   Changes: Text improved; terminology changed; formulas and symbols adopted.

   In fact, when the lake continues to increase, lake evaporation increases and an equilibrium between input and output will be reached. Therefore, lake water level would not rise by 400 m over the basin within only 10 ka, even there is a positive long-

290    term mean annual water balance.

   Reply: See my comments above. Since lake evaporation is included in the water balance, a positive water balance would in fact result in an increase in terrestrial water storage (TWS) and subsequent recharge of the reservoirs including the lakes as stated in P8 L227-231. As proxy-based studies have shown, lake levels in the Qaidam basin have indeed shown drastic changes in the past during the Pleistocene glacial cycles. The interglacial periods have usually been shorter than the glacial periods,

295    lasting for about 10 to 15 ka. So, the question arises how the mega-lake system could have been restored, at least partially, during such comparably short time periods. The findings of my study imply that lake restoration could take place within such short time periods without requiring drastic climatic changes, for which there is no evidence. I do not say that the sensitivity of the water balance of the Qaidam basin has been the same as today. But as Dowsett et al. (2010); cited in the manuscript, have stated that the paleogeographic situation of the mid-Pliocene epoch has been similar to the presence, this holds even more

300    true for the Pleistocene, and definitely true for the near future. I am not aware of any study that has shown that the sensitivity of the water balance to changes in air temperature and humidity has been different than today. So far, this remains to be an unproven hypothesis. Therefore, I do not understand, why my approach, i.e., applying the physically based sensitivities as

shown in Table 2 to estimates of the plausible range of changes in air temperature and precipitation during the mid-Pliocene as inferred from independent studies, both proxy-based and model results, should be scientifically wrong.

Changes: Text improved; feedbacks better discussed.

2. Line 166-174, as the author said, both lake area and lake number in the Qaidam basin have increased from the last two decades. The groundwater in the Qaidam basin was recharged between 2003 and 2012 due to changes in terrestrial water storage of 20.6 km3 at rate of 8 mm/a. These studies indicate that there should be considerable positive water balance during the last two decades in Qaidam basin, which contrasts with the main conclusion of the study, which shows the water balance of the basin is almost zero (Line 174).

Reply: I discussed that the results from GRACE satellite-based analyses are also showing that the hyperaridity of the low-lying regions of the Qaidam basin is misleading, when water balance of the entire basin including the high-altitude regions is to be assessed. For the same time period, the GRACE data show +8 mm/a while the HAR data set indicates 0 mm/a. Unfortunately, I could not find a specification of the error of the GRACE-based results in the publication of Jiao et al. (2015), but it is obvious that the error is in the order of at least ±10 mm/a. The uncertainty of the water balance as derived from the HAR data set is ±34 mm/a for the entire study period as specified in P4 L94. So, both results are not in contradiction. I also discussed further results from independent studies that clearly indicate that the water balance of the Qaidam basin has been positive during this period which showed warmer and less dry climate conditions. There is no study that is in contradiction to the results derived from the HAR data set for the recent time period.

Changes: Text improved; further references added.

3. L194-197: "the estimated changes in precipitation due to changes in air temperature would be 52 to 105 mm/a. The mean change in water balance as inferred from the changes in air temperature would lead to a positive mean annual water balance of 49 mm/a." I cannot believe this derivation. The positive water balance is too big, because most of precipitation would be lost through evaporation in such a dry environment.

Reply: It is not a matter of belief but just the result of a computation using a) the present-day sensitivity of mean annual precipitation to changes in mean annual air temperature, and b) the present-day sensitivity of mean annual water balance to changes in mean annual precipitation, all quantities spatially averaged over the entire Qaidam basin, i.e., including the less dry and colder high-mountain regions. The starting point of this computation is the range of plausible changes in mean annual air temperature relative to the present-day situation, which has been taken from the literature, and is conservatively set to 1 to 2 K in my study. Applying a) gives 52 to 102 mm/a more presentation than today. This range of precipitation change is in accordance with the paleoclimate studies I have cited in the manuscript. Taking these values as input in b) leads to 42 to 84 mm/a higher values for the water balance than today. In absolute terms, i.e., taking the value of -14 mm/a as the current estimate, the water balance would be between +28 and + 70 mm/a, i.e., +49 mm/a on average. Please regard that any kind of evaporative water losses are in fact included in the sensitivity of b).

Changes: Text improved; sensitivity study better explained; feedbacks better discussed.

4. The main conclusion of the study is 'near-future climates not much different from present conditions could cause rising lake levels and expanding lake areas, and may result in restoration of the Qaidam mega-lake system over geological time scales' (line12-14). Although I am not a paleo-climatologist, I guess the formation of megalake system over geological time scales might have underwent special climate conditions (e.g. different scales of lake-air interaction, different land cover condition), and this speculation is quite uncertain. Even current climate yields positive mass balance, this positive balance would be lost soon due to the increase of lake evaporation when lake expands.

Reply: As discussed above, my study results are neither a reconstruction of the water balance of the mid-Pliocene nor a projection of the future water balance. I only conclude that there is no need for drastic climatic or environmental changes to explain the existence of a mega-lake system in the Qaidam basin. Using the results from the scientific literature that I have referenced in the manuscript, there is proven evidence that the mega-lake system has actually existed in the Pliocene. So, there must have been a physical mechanism that has allowed the mega-lake system to sustain such a large lake surface (about 120.000 km²) for long time without losing more water by lake evaporation and evapotranspiration over land than gaining by precipitation. The feedback mechanism that limits lake growth has been prominently mentioned in the conclusions part of my manuscript to make clear that the positive water balance will not be able to persist over unlimited time, and finally must reach an equilibrium. However, I stated that we do not yet have enough knowledge on this mechanism to use it for quantifying the effect, which would be prerequisite for any attempt to reconstruct the paleo-lake evolution in the Qaidam basin.

Changes: Text improved; further references added; text parts shifted from the Conclusions to the Discussion section.

**Replies to Reviewer 3**

The paper provides a nice straightforward estimation of the regional water balance of the Qaidam basin. It is interesting to observe that the water balance calculations confirm that the balance is near zero. It implies that although parts of the basin has hyper arid conditions, there is apparently a zero balance. This as such answers the question: can a lake exist for prolonged periods in this basin under arid conditions.

Reply: This is exactly the main focus of the study. I will need to make this much better visible than in the current version of the manuscript.

Changes: Text improved.

I am struggling with the way this is framed. We have no future time series so we can only speculate what will happen in the future. It is interesting that the current trends of climate change seems to lead to a positive water balance. The problem I have is that this does not automatically imply that the lake level will rise.

Reply: Any kind of projection, e.g. the climate projections used by the IPCC or studies of future changes of the Asian monsoon system, would then be purely speculative. As the formulations in the first paragraph of the Conclusions section (lines 223-231)

clearly indicates, I don't make any predictions but only refer to the consequences of a slightly positive water balance. All sentences are written as conditional statements, which means that I am fully aware that we don't know how the situation of the basin's water balance will develop in the future. If the long-term mean water balance of the Qaidam basin would be positive, then (and only then) the consequence would be that reservoirs would start to restore. I described this in the first paragraph, and I also stated that the first response would be that the groundwater reservoirs would be recharged, and then the lakes would also start to restore, which seems to be happening already now (see literature review in lines 162-174).

Changes: Text improved; further references added.

I think the claim of a tipping point (line 15) is not substantiated, it could be a threshold but a tipping point suggest a complete new system equilibrium. This can not be predicted with linear regressions based on the current system dynamics.

Reply: That is correct; I will use the term "threshold" in a revised version of the manuscript.

Changes: Term "tipping point" replaced by "threshold".

The link with the Mega Lake is also unclear. No information of its extent, depth etc is given. The existence of such a lake would invalidate many of the assumptions now made to calculate the water balance.

Reply: There is no spatially explicit information on the mega-lake system existing from the literature. But the geological evidence of its existence is referenced in the manuscript. The intention of the manuscript is not to reconstruct the mid-Pleistocene conditions of the mega-lake system but only to find a physical explanation for the existence of such a mega-lake system. The main answer is that the high-mountain parts of the Qaidam basin could have acted as regional water towers under slightly wetter and warmer climates that prevailed during the mid-Pliocene epoch (as indicated by the literature referenced in my study) such that the long-term water balance would not have been negative. In an (unknown) equilibrium state, water balance would have been zero, and positive during time periods when the mega-lake system would have been restored after periods of lake shrinkage, which have occurred several times in geological history.

Changes: Text improved.

Please discuss possible feedbacks in the studied system. You now seem to assume they dot not exist, which I consider unlikely.

Reply: In the last paragraph of the Conclusions section (lines 241-246) I discussed feedbacks that need to be addressed in future studies. I mentioned the most important negative feedback, i.e., increasing lake evaporation, that would finally lead to zero water balance even when precipitation would be higher than today due to changes in atmospheric circulation. There must be an upper, yet unknown limit for lake growth, which is, however, not subject of this study. I will make this point even more explicit in a revised version of the manuscript. I also mentioned the feedback that local recycling of water could be intensified when lake surface increases. Higher lake evaporation could potentially result in more precipitation within the boundaries of the Qaidam basin, which is large enough to support local recycling. This is, however, not yet quantifiable. I also discussed

feedbacks due to orographic changes (see replies below), which would be relevant for the mid-Pliocene epoch but not for the next hundred thousand years.

Changes: Text improved; feedbacks better discussed.

Minor comments:

Line 7: the phrase increasingly arid climates is confusing given the fact that the analysis is not suitable to deal with large long term climate changes.

Reply: This sentence in the abstract is referring to the scientific literature that forms the motivation and background of my study. As it is part of the abstract, no references to scientific studies are given there. This is done in the Introduction section.

Changes: None.

Line 14: the restoration of the mage-lake is very speculative given the fact that the calculations are based on 14 years of data. Such a restoration would need ten thousands of years.

Reply: See my replies above. I don't speculate on the future development of the basin but only describe what would happen if long-term mean water balance would become positive, which is a realistic scenario as my study reveals. My study shows that this could happen (or may even have already started to happen) under subtle changes in the regional atmospheric circulation. Using the results from my study, I made an example calculation (lines 223-231) to illustrate that even a slightly positive, physically realistic water balance would have a huge cumulative effect over ten thousand years. This is, of course, not a prediction since no feedbacks are considered in the example calculation, and future changes in atmospheric circulation are not known!

Changes: Text improved.

Line 24: the disappearance of the lake during the last 100 ka is intriguing. What was different?

Reply: This is a very interesting question but not part of my study. The disappearance of the mega-lake system is described in the scientific literature. However, the details of the lake and climate conditions are not well constrained by observational data. I would regard to find an answer to this question as a highly ambitious but valuable goal for future research. My study reveals that climates colder and drier than today would probably drop the water balance due to reduction in precipitation. Although the mega-lake system disappeared there are nevertheless substantial variations in lake levels during this period. Such lake-level variations have been reported even for the Holocene, showing that the remaining lakes are highly sensitive to small changes in climate conditions, as revealed in my study for present-day geographic conditions.

Changes: Text improved.

Line 30-40: please provide units and support (10*10 km for example)

Reply: I will need to reformulate the manuscript regarding the terms "net precipitation" and "water balance", as already mentioned in my replies to the other reviewers. The term "water balance shall only be used when net precipitation is integrated over the entire basin. I will add suitable references as support. However, I don't understand, in which context units are missing.

Changes: Text improved; terminology changed; formulas and symbols adopted; information on units added.

Line 45-50: please formulate a concrete testable hypothesis. I propose that within the Qiadam basin the water balance is near zero.

Reply: In fact, this was not the starting point of my study. I didn't expect a near-zero water balance. Moreover, the original hypothesis was that increasing air temperatures would make the water balance less negative under present-day conditions. The results of my study showed that this is indeed happening, but the statistical analysis showed that not air temperate but specific humidity and the combined effect of air temperature and specific humidity are driving annual water balance variations. I will explicitly formulate this in a revised version of the manuscript.

Changes: Text improved; hypotheses explicitly formulated.

Line 74: what are valid data? How is this supported by evidence?

Reply: The expression "valid data" refers to the fact that meteorological data are existing for the time period and are not flagged as invalid by the data provider.

Changes: Text improved.

Line 82-84: is there possibly spatial autocorrelation in your analysis?

Reply: The results presented in Figure 4 are spatially averaged pairs of annual values for the entire Qaidam basin, thus there are no problems with spatial autocorrelation.

Changes: None.

Line 103: are you implying that there are feedbacks in the system? If this is the case, it implies that you should be very carful to use your regression relationship outside current conditions.

Reply: There are definitely feedbacks in the system when paleographic or future geographic conditions (especially total lake surface) would change. For this reason, I always use the term "sensitivity" and refer to the present-day geographic conditions. All statements referring to the mid-Pleistocene epoch or to the future are thus conditional, indicating that they are only simplified projections, and shall in no way be regarded as predictions.

Changes: Text improved; feedbacks better discussed.

Line 120: an important conclusion. Please emphasize this more in abstract and conclusions

Reply: I will consider this in a revised version of the manuscript.

Changes: Text improved.

Line 134: same as above important insight in current system

Reply: I will consider this in a revised version of the manuscript.

475    Changes: Text improved.

Line 174: how sensitive is your study for spatial resolution?

Reply: Interestingly and surprisingly, the results did not substantially differ between the 30 km and 10 km gridded HAR data (see lines 218-221 and the results presented in the supplement). As long as major features of the orography of the Qaidam

480    basin are present in the dynamical downscaling runs, and thus, the mesoscale atmospheric processes induced by orography are resolved by the model, then the results are similar. I will add more details on this topic in a revised version of the manuscript (see also my reply below).

Changes: Text improved; HAR 30 km results better explained.

485    Line 197: how long would it take to fill up the whole mage-lake (with this rate of 49 mm/a).

Reply: I have made an example calculation using 40 mm/a as long-term mean value for the water balance (see lines 227-231). Since no feedbacks are included and no differentiation between the reservoirs (groundwater, lakes, …) has been made, the result (400 m rise of the water table in any of the reservoirs averaged over the entire basin within ten thousand years) is only illustrating that huge lakes could form within a geologically short time period. This calculation serves to show that 'rapid'

490    (time scales of thousands to hundred thousand years) lake-level changes as reported in the scientific literature are not physically unfeasible. Since there is no spatially explicit information on area, depth or volume of the mega-lake system available from the literature, a spatially explicit computation would be impossible, even when feedbacks would be included (making the water balance time dependent).

Changes: Text improved.

495

Line 238: how important can the orographic precipitation be? You now assume this not to happen. Ad how does the presence of a big lake affect this effect?

Reply: My study shows that orographic precipitation is one of the most important processes with respect to basin-wide water balance. As Figure 3 shows and is discussed in the manuscript (see e.g. lines 111-115), precipitation increases with altitude,

500    which is a direct consequence of orographic precipitation. It is not altitude itself but relief that induces precipitation in the mountain ranges. The results of my study only show the sensitivity of water-balance components to variations of climate conditions under present-day geographic conditions. A changed orography and the existence of a mega-lake system would induce additional feedbacks (as discussed in my manuscript; see my replies above). This could also affect orographic precipitation. The effect of lower altitudes is partly captured by the HAR 30 km data set, since highest altitudes in the 30 km

gridded topography are lower by a few hundred meters than in the HAR 10 km data set (see lines 220-221). This has two counteracting effects as discussed in the manuscript (see lines 210-221): precipitation would tend to decrease since orographic precipitation would possibly be weakened, but blocking of atmospheric water transport to the Qaidam basin by high mountain ranges would also be reduced. The latter effect seems to be stronger than the first one, since mean water balance of the Qaidam basin is +3 mm/a in the HAR 30 km data set while it amounts to -14 mm/a in the HAR 10 km data set.

Changes: Text improved; HAR 30 km results better explained.

[revised manuscript text omitted]
(P\text{-}ET)$ min | $\Delta(P\text{-}ET)$ max | $P\text{-}ET$ min | $P\text{-}ET$ max | $P\text{-}ET$ mean |
|---|---|---|---|---|---|---|---|---|---|---|---|---|---|
| $\partial q/\partial T$ | 0.288 | g/(kg·K) | 1.0 | 2.0 | 0.3 | 0.6 | | | | | | | |
| $\partial P/\partial T$ | 52.40 | mm/(a·K) | | | | | 52 | 105 | | | | | |
| $\partial(P\text{-}ET)/\partial T$ | 42.17 | mm/(a·K) | | | | | | | 42 | 84 | 28 | 70 | 49 |
| $\partial P/\partial q$ | 185.6 | mm·kg/(a·g) | | | 0.3 | 0.6 | 53 | 107 | | | | | |
| $\partial(P\text{-}ET)/\partial q$ | 139.7 | mm·kg/(a·g) | | | | | | | 40 | 81 | 26 | 67 | 46 |
| $\partial(P\text{-}ET)/\partial P$ | 0.725 | mm/mm | | | | | 50 | 100 | 36 | 73 | 22 | 59 | 40 |

Supplement to

**Survival of the Qaidam Mega-Lake System under Mid-Pliocene Climates and its Restoration under Future Climates**

Dieter Scherer[1]

5    [1] Chair of Climatology, Technische Universität Berlin, Berlin, 12165, Germany

*Correspondence to*: Dieter Scherer (dieter.scherer@tu-berlin.de)

**1 Supplementary Figures**

[Figure]

[Figure]

10    **Figure S1: Mean annual air temperature (2 m above ground)**  during **the study period of 14 hydrological years (2001-2014) for the Qaidam** **B**asin **(QB) and its surrounding regions derived from the HAR 10 km data set**. **Black line: boundary of the Q****B (Lehner and Grill, 2013). Topographic shading is based on DEM data from the SRTM. Black dots indicate the locations of the eight GSOD stations within or nearby the QB.**

[Figure]

[Figure]

**Figure S2:** Mean annual specific humidity (2 m above ground) during the study period of 14 hydrological years (2001-2014) for the Qaidam Basin (QB) and its surrounding regions derived from the HAR 10 km data set. Black line: boundary of the QB (Lehner and Grill, 2013). Topographic shading is based on DEM data from the SRTM. Black dots indicate the locations of the eight GSOD stations within or nearby the QB.

[Figure]

[Figure]

**Figure S3:** Mean annual precipitation during the study period of 14 hydrological years (2001-2014) for the Qaidam Basin (QB) and its surrounding regions derived from the HAR 10 km data set. Black line: boundary of the QB (Lehner and Grill, 2013). Topographic shading is based on DEM data from the SRTM. Black dots indicate the locations of the eight GSOD stations within or nearby the QB.

[Figure]

Qaidam Basin: mean annual rainfall (HAR 10 km)

[Figure]

Qaidam HAR V1 (10 km): long−term mean annual rainfall

**Figure S4: Mean annual rainfall** during **the study period of 14 hydrological years (2001-2014) for the Qaidam** B**asin** (QB) **and its surrounding regions** derived from the HAR 10 km data set**. Black line: boundary of the Q**B **(Lehner and Grill, 2013). Topographic shading is based on DEM data from the SRTM.** Black dots indicate the locations of the eight GSOD stations within or nearby the QB.

[Figure]

[Figure]

**Figure S5: Mean annual snowfall during the study period of 14 hydrological years (2001-2014) for the Qaidam Basin (QB) and its surrounding regions derived from the HAR 10 km data set. Black line: boundary of the QB (Lehner and Grill, 2013). Topographic shading is based on DEM data from the SRTM. Black dots indicate the locations of the eight GSOD stations within or nearby the QB.**

[Figure]

[Figure]

40 **Figure S6: Mean annual actual evapotranspiration** during **the study period of 14 hydrological years (2001-2014) for the Qaidam** **B**asin **(QB)** **and its surrounding regions** _derived from the HAR 10 km data set_**. Black line: boundary of the Q****B (Lehner and Grill, 2013). Topographic shading is based on DEM data from the SRTM.** _Black dots indicate the locations of the eight GSOD stations within or nearby the QB._

[Figure]

[Figure]

**Figure S7:** Mean annual net precipitation, i.e., precipitation *P* minus actual evapotranspiration *ET*, during the study period of 14 hydrological years (2001-2014) for the Qaidam Basin (QB) and its surrounding regions derived from the HAR 10 km data set. Black line: boundary of the QB (Lehner and Grill, 2013). Topographic shading is based on DEM data from the SRTM. Black dots indicate the locations of the eight GSOD stations within or nearby the QB.

[Figure]

**Figure S8: Specific humidity *q* air temperature *T* the Qaidam Basin (QB) during the hydrological years 2001 to 2014 derived from the HAR 10 km data set. Dotted lines: mean annual values; solid lines: regression lines; light grey shades: confidence interval; dashed lines: prediction interval.**

[Figure]

**Figure S9: Annual actual evapotranspiration *ET* determined by the SEBS-based study of Jin et al. (2013) versus *ET* of the HAR  km data set in the Qaidam Basin (QB) for the calendar years 2001-2011 (left panel) and 2005-2011 (right panel). Dotted lines: mean annual values; solid lines: regression lines; light grey shades: confidence intervals; dashed lines: prediction intervals.**

[Figure]

[Figure]

65    **Figure S10: Annual precipitation**  **(upper left panel), actual evapotranspiration**  **(upper right panel), and water balance**
 **(lower panel) in the Qaidam** **Basin (QB) during the hydrological years 2001 to 2014 as in Fig. 2 but derived from the HAR**
**30 km**  **data set. Upper left panel: light grey bars: annual snowfall** **; dark grey bars: annual**
**rainfall** **. Dotted lines: mean annual values.**

[Figure]

[Figure]

**Figure S11:** Water balance  versus precipitation  (upper left panel) and actual evapotranspiration  (upper right panel); precipitation versus air temperature  (middle left panel) and specific humidity  (middle right panel);  water balance versus air temperature (lower left panel) and specific humidity  (lower right panel) in the Qaidam Basin (QB) during the hydrological years 2001 to 2014 as in Fig 4 but derived from the HAR 30 km  data set. Dotted

lines: mean annual values; solid lines: regression lines; light grey shades: confidence intervals; dashed lines: prediction intervals.

**2 Supplementary Tables**

**Table S2: Monthly and annual air temperature *T* (in deg C)  in the Qaidam bBasin (QB) for the 14 hydrological years (2001-2014) covered by the HAR 10 km data set.**

| *T* | 10 | 11 | 12 | 1 | 2 | 3 | 4 | 5 | 6 | 7 | 8 | 9 | year |
|---|---|---|---|---|---|---|---|---|---|---|---|---|---|
| 2001 | -2.0 | -10.3 | -14.2 | -16.1 | -11.5 | -7.7 | -1.8 | 3.5 | 9.4 | 13.3 | 10.6 | 6.5 | **-1.6** |
| 2002 | -1.5 | -8.3 | -14.5 | -16.7 | -10.8 | -6.6 | 1.0 | 4.1 | 9.9 | 12.4 | 11.6 | 5.1 | **-1.2** |
| 2003 | -3.1 | -9.5 | -12.3 | -13.2 | -10.2 | -5.9 | 0.2 | 3.9 | 9.2 | 11.2 | 10.6 | 6.1 | **-1.0** |
| 2004 | -1.2 | -7.4 | -12.7 | -14.9 | -11.6 | -4.3 | 1.4 | 4.1 | 8.4 | 11.2 | 10.3 | 5.3 | **-0.9** |
| 2005 | -2.0 | -10.6 | -11.8 | -13.1 | -10.4 | -4.8 | -0.5 | 5.1 | 9.6 | 11.6 | 10.9 | 7.3 | **-0.7** |
| 2006 | -0.9 | -8.7 | -13.4 | -11.5 | -8.5 | -6.6 | -0.7 | 5.1 | 9.7 | 13.0 | 12.9 | 7.7 | **-0.1** |
| 2007 | 0.8 | -7.3 | -13.7 | -16.2 | -10.0 | -5.8 | 0.9 | 7.0 | 9.4 | 11.5 | 12.0 | 6.6 | **-0.4** |
| 2008 | -0.6 | -7.2 | -11.7 | -14.8 | -14.9 | -5.4 | 0.1 | 7.1 | 10.0 | 11.7 | 9.8 | 7.2 | **-0.7** |
| 2009 | 0.3 | -7.3 | -11.0 | -12.8 | -8.4 | -5.0 | 2.6 | 5.3 | 10.2 | 12.1 | 10.7 | 8.0 | **0.4** |
| 2010 | -0.6 | -8.2 | -12.6 | -11.5 | -8.9 | -5.3 | -0.8 | 5.4 | 9.8 | 13.9 | 12.8 | 7.7 | **0.2** |
| 2011 | -0.2 | -8.1 | -14.3 | -16.5 | -8.8 | -7.7 | 0.8 | 6.4 | 10.7 | 12.9 | 12.8 | 7.7 | **-0.3** |
| 2012 | 0.9 | -5.6 | -12.3 | -17.9 | -11.9 | -6.8 | 0.1 | 6.4 | 10.7 | 12.7 | 12.6 | 7.6 | **-0.3** |
| 2013 | -1.0 | -9.1 | -13.7 | -15.2 | -9.6 | -2.5 | 1.3 | 6.3 | 11.5 | 12.9 | 13.8 | 7.4 | **0.2** |
| 2014 | 0.8 | -10.3 | -15.2 | -14.1 | -11.3 | -4.4 | -0.1 | 5.0 | 10.2 | 13.4 | 11.6 | 7.9 | **-0.5** |
| **mean** | **-0.7** | **-8.4** | **-13.1** | **-14.6** | **-10.5** | **-5.6** | **0.3** | **5.3** | **9.9** | **12.4** | **11.6** | **7.0** | **-0.5** |

**Table S2: Monthly and annual specific humidity  q (in g/kg) in the Qaidam Basin (QB) during the 14 hydrological years (2001-2014) covered by the HAR  (10 km data set.**

| q | 10 | 11 | 12 | 1 | 2 | 3 | 4 | 5 | 6 | 7 | 8 | 9 | year |
|---|----|----|----|---|---|---|---|---|---|---|---|---|------|
| 2001 | 1.9 | 1.2 | 1.0 | 0.8 | 0.9 | 1.0 | 1.8 | 2.3 | 3.1 | 4.2 | 4.7 | 3.8 | **2.2** |
| 2002 | 1.7 | 1.3 | 1.0 | 0.9 | 1.1 | 1.4 | 2.2 | 2.7 | 4.0 | 5.5 | 4.4 | 4.0 | **2.5** |
| 2003 | 2.0 | 1.4 | 1.3 | 1.1 | 1.4 | 1.7 | 2.4 | 2.5 | 3.5 | 4.0 | 4.3 | 3.1 | **2.4** |
| 2004 | 1.8 | 1.6 | 1.1 | 1.1 | 1.3 | 1.7 | 1.9 | 2.7 | 3.3 | 4.5 | 5.0 | 2.9 | **2.4** |
| 2005 | 2.0 | 1.4 | 1.3 | 1.1 | 1.2 | 1.9 | 1.9 | 2.8 | 4.2 | 6.2 | 5.7 | 4.0 | **2.8** |
| 2006 | 2.0 | 1.2 | 0.8 | 1.1 | 1.5 | 1.1 | 1.7 | 2.4 | 4.3 | 6.4 | 5.1 | 3.5 | **2.6** |
| 2007 | 2.2 | 1.5 | 1.0 | 0.7 | 1.2 | 1.7 | 2.0 | 2.5 | 4.4 | 5.5 | 5.9 | 4.3 | **2.8** |
| 2008 | 2.7 | 1.3 | 1.0 | 1.0 | 1.1 | 1.5 | 1.8 | 3.0 | 4.2 | 6.1 | 4.6 | 4.5 | **2.7** |
| 2009 | 2.6 | 1.6 | 1.2 | 1.1 | 1.3 | 1.5 | 2.0 | 2.9 | 4.1 | 6.1 | 4.8 | 5.3 | **2.9** |
| 2010 | 2.4 | 1.4 | 1.1 | 1.1 | 1.4 | 1.7 | 2.0 | 3.1 | 5.6 | 7.1 | 5.1 | 4.7 | **3.1** |
| 2011 | 2.6 | 1.3 | 0.9 | 0.7 | 1.3 | 1.3 | 2.0 | 3.2 | 4.8 | 5.4 | 5.6 | 4.0 | **2.8** |
| 2012 | 2.3 | 1.7 | 0.9 | 0.8 | 1.2 | 1.5 | 1.7 | 3.4 | 5.0 | 6.5 | 6.0 | 3.1 | **2.8** |
| 2013 | 1.9 | 1.1 | 0.9 | 0.9 | 1.2 | 1.2 | 1.6 | 3.2 | 4.5 | 5.9 | 5.2 | 3.3 | **2.6** |
| 2014 | 2.0 | 1.2 | 0.9 | 0.8 | 1.1 | 1.5 | 2.0 | 2.1 | 4.6 | 5.4 | 5.0 | 4.2 | **2.6** |
| **mean** | **2.2** | **1.4** | **1.0** | **0.9** | **1.2** | **1.5** | **1.9** | **2.8** | **4.3** | **5.6** | **5.1** | **3.9** | 2.7 |

| q (g/kg) | 10 | 11 | 12 | 1 | 2 | 3 | 4 | 5 | 6 | 7 | 8 | 9 | year |
|---|----|----|----|---|---|---|---|---|---|---|---|---|------|
| 2001 | 1.9 | 1.2 | 1.0 | 0.8 | 0.9 | 1.0 | 1.8 | 2.3 | 3.1 | 4.2 | 4.7 | 3.8 | **2.2** |
| 2002 | 1.7 | 1.3 | 1.0 | 0.9 | 1.1 | 1.4 | 2.2 | 2.7 | 4.0 | 5.5 | 4.4 | 4.0 | **2.5** |
| 2003 | 2.0 | 1.4 | 1.3 | 1.1 | 1.4 | 1.7 | 2.4 | 2.5 | 3.5 | 4.0 | 4.3 | 3.1 | **2.4** |
| 2004 | 1.8 | 1.6 | 1.1 | 1.1 | 1.3 | 1.7 | 1.9 | 2.7 | 3.3 | 4.5 | 5.0 | 2.9 | **2.4** |
| 2005 | 2.0 | 1.4 | 1.3 | 1.1 | 1.2 | 1.9 | 1.9 | 2.8 | 4.2 | 6.2 | 5.7 | 4.0 | **2.8** |
| 2006 | 2.0 | 1.2 | 0.8 | 1.1 | 1.5 | 1.1 | 1.7 | 2.4 | 4.3 | 6.4 | 5.1 | 3.5 | **2.6** |
| 2007 | 2.2 | 1.5 | 1.0 | 0.7 | 1.2 | 1.7 | 2.0 | 2.5 | 4.4 | 5.5 | 5.9 | 4.3 | **2.8** |
| 2008 | 2.7 | 1.3 | 1.0 | 1.0 | 1.1 | 1.5 | 1.8 | 3.0 | 4.2 | 6.1 | 4.6 | 4.5 | **2.7** |
| 2009 | 2.6 | 1.6 | 1.2 | 1.1 | 1.3 | 1.5 | 2.0 | 2.9 | 4.1 | 6.1 | 4.8 | 5.3 | **2.9** |
| 2010 | 2.4 | 1.4 | 1.1 | 1.1 | 1.4 | 1.7 | 2.0 | 3.1 | 5.6 | 7.1 | 5.1 | 4.7 | **3.1** |
| 2011 | 2.6 | 1.3 | 0.9 | 0.7 | 1.3 | 1.3 | 2.0 | 3.2 | 4.8 | 5.4 | 5.6 | 4.0 | **2.8** |
| 2012 | 2.3 | 1.7 | 0.9 | 0.8 | 1.2 | 1.5 | 1.7 | 3.4 | 5.0 | 6.5 | 6.0 | 3.1 | **2.8** |
| 2013 | 1.9 | 1.1 | 0.9 | 0.9 | 1.2 | 1.2 | 1.6 | 3.2 | 4.5 | 5.9 | 5.2 | 3.3 | **2.6** |
| 2014 | 2.0 | 1.2 | 0.9 | 0.8 | 1.1 | 1.5 | 2.0 | 2.1 | 4.6 | 5.4 | 5.0 | 4.2 | **2.6** |
| **mean** | **2.2** | **1.4** | **1.0** | **0.9** | **1.2** | **1.5** | **1.9** | **2.8** | **4.3** | **5.6** | **5.1** | **3.9** | 2.7 |

85

**Table S3: Monthly and annual precipitation *P* (in mm/month or mm/a) *P* in the Qaidam bBasin (QB) forduring the 14 hydrological years (2001-2014) covered by the HAR V1 (10 km) data set.**

| *P* | 10 | 11 | 12 | 1 | 2 | 3 | 4 | 5 | 6 | 7 | 8 | 9 | year |
|------|----|----|----|----|----|----|----|----|----|----|----|----|------|
| 2001 | 5 | 4 | 6 | 3 | 3 | 3 | 19 | 16 | 17 | 11 | 20 | 15 | **122** |
| 2002 | 2 | 1 | 6 | 3 | 3 | 10 | 16 | 25 | 34 | 25 | 12 | 22 | **160** |
| 2003 | 5 | 3 | 6 | 4 | 8 | 17 | 28 | 23 | 19 | 21 | 26 | 10 | **171** |
| 2004 | 3 | 5 | 5 | 8 | 10 | 11 | 12 | 29 | 24 | 24 | 24 | 8 | **163** |
| 2005 | 6 | 4 | 8 | 3 | 7 | 13 | 9 | 27 | 29 | 55 | 39 | 26 | **229** |
| 2006 | 5 | 2 | 1 | 3 | 12 | 4 | 16 | 16 | 37 | 39 | 23 | 12 | **170** |
| 2007 | 8 | 3 | 2 | 1 | 5 | 14 | 14 | 20 | 59 | 35 | 28 | 33 | **222** |
| 2008 | 7 | 2 | 1 | 6 | 6 | 7 | 13 | 24 | 26 | 60 | 25 | 35 | **212** |
| 2009 | 13 | 5 | 6 | 6 | 9 | 14 | 11 | 44 | 20 | 57 | 33 | 38 | **256** |
| 2010 | 12 | 5 | 4 | 5 | 7 | 21 | 14 | 39 | 77 | 42 | 18 | 26 | **270** |
| 2011 | 11 | 1 | 3 | 3 | 6 | 12 | 14 | 35 | 49 | 29 | 35 | 22 | **221** |
| 2012 | 7 | 5 | 1 | 3 | 6 | 10 | 10 | 44 | 46 | 60 | 40 | 14 | **246** |
| 2013 | 12 | 4 | 4 | 3 | 7 | 2 | 7 | 30 | 24 | 41 | 18 | 23 | **174** |
| 2014 | 5 | 3 | 2 | 1 | 5 | 9 | 19 | 14 | 41 | 31 | 27 | 30 | **188** |
| mean | **7** | **3** | **4** | **4** | **7** | **11** | **15** | **28** | **36** | **38** | **26** | **23** | **200** |

| *P* (mm) | 10 | 11 | 12 | 1 | 2 | 3 | 4 | 5 | 6 | 7 | 8 | 9 | year |
|----------|----|----|----|----|----|----|----|----|----|----|----|----|------|
| 2001 | 5 | 4 | 6 | 3 | 3 | 3 | 19 | 16 | 17 | 11 | 20 | 15 | **122** |
| 2002 | 2 | 1 | 6 | 3 | 3 | 10 | 16 | 25 | 34 | 25 | 12 | 22 | **160** |
| 2003 | 5 | 3 | 6 | 4 | 8 | 17 | 28 | 23 | 19 | 21 | 26 | 10 | **171** |
| 2004 | 3 | 5 | 5 | 8 | 10 | 11 | 12 | 29 | 24 | 24 | 24 | 8 | **163** |
| 2005 | 6 | 4 | 8 | 3 | 7 | 13 | 9 | 27 | 29 | 55 | 39 | 26 | **229** |
| 2006 | 5 | 2 | 1 | 3 | 12 | 4 | 16 | 16 | 37 | 39 | 23 | 12 | **170** |
| 2007 | 8 | 3 | 2 | 1 | 5 | 14 | 14 | 20 | 59 | 35 | 28 | 33 | **222** |
| 2008 | 7 | 2 | 1 | 6 | 6 | 7 | 13 | 24 | 26 | 60 | 25 | 35 | **212** |
| 2009 | 13 | 5 | 6 | 6 | 9 | 14 | 11 | 44 | 20 | 57 | 33 | 38 | **256** |
| 2010 | 12 | 5 | 4 | 5 | 7 | 21 | 14 | 39 | 77 | 42 | 18 | 26 | **270** |
| 2011 | 11 | 1 | 3 | 3 | 6 | 12 | 14 | 35 | 49 | 29 | 35 | 22 | **221** |
| 2012 | 7 | 5 | 1 | 3 | 6 | 10 | 10 | 44 | 46 | 60 | 40 | 14 | **246** |
| 2013 | 12 | 4 | 4 | 3 | 7 | 2 | 7 | 30 | 24 | 41 | 18 | 23 | **174** |
| 2014 | 5 | 3 | 2 | 1 | 5 | 9 | 19 | 14 | 41 | 31 | 27 | 30 | **188** |
| mean | **7** | **3** | **4** | **4** | **7** | **11** | **15** | **28** | **36** | **38** | **26** | **23** | **200** |

90

**Table S4: Monthly and annual rainfall $P_{rain}$ (in mm/month or mm/a) $P_{rain}$ in the Qaidam bBasin (QB) forduring the 14 hydrological years (2001-2014) covered by the HAR V1 (10 km) data set.**

| $P_{rain}$ | 10 | 11 | 12 | 1 | 2 | 3 | 4 | 5 | 6 | 7 | 8 | 9 | year |
|---|---|---|---|---|---|---|---|---|---|---|---|---|---|
| 2001 | 1 | 0 | 0 | 0 | 0 | 1 | 3 | 3 | 7 | 10 | 13 | 8 | **45** |
| 2002 | 0 | 0 | 0 | 0 | 0 | 0 | 4 | 8 | 19 | 19 | 9 | 10 | **70** |
| 2003 | 0 | 0 | 0 | 0 | 0 | 1 | 7 | 5 | 8 | 12 | 15 | 3 | **52** |
| 2004 | 0 | 0 | 0 | 0 | 0 | 1 | 3 | 7 | 14 | 16 | 18 | 3 | **62** |
| 2005 | 0 | 0 | 0 | 0 | 1 | 1 | 2 | 8 | 15 | 42 | 28 | 12 | **110** |
| 2006 | 0 | 0 | 0 | 0 | 0 | 0 | 5 | 5 | 19 | 36 | 16 | 6 | **87** |
| 2007 | 2 | 1 | 0 | 0 | 0 | 1 | 5 | 5 | 31 | 26 | 24 | 16 | **110** |
| 2008 | 1 | 0 | 0 | 0 | 0 | 1 | 3 | 10 | 16 | 46 | 15 | 19 | **112** |
| 2009 | 3 | 0 | 0 | 0 | 0 | 1 | 3 | 13 | 11 | 44 | 23 | 27 | **126** |
| 2010 | 1 | 0 | 0 | 0 | 1 | 1 | 2 | 14 | 54 | 36 | 14 | 17 | **141** |
| 2011 | 1 | 0 | 0 | 0 | 1 | 1 | 3 | 10 | 33 | 22 | 30 | 12 | **113** |
| 2012 | 2 | 1 | 0 | 0 | 0 | 0 | 2 | 17 | 28 | 51 | 35 | 5 | **142** |
| 2013 | 2 | 0 | 0 | 0 | 0 | 1 | 2 | 11 | 18 | 33 | 16 | 9 | **93** |
| 2014 | 1 | 0 | 0 | 0 | 0 | 1 | 4 | 4 | 24 | 26 | 20 | 14 | **95** |
| **mean** | **1** | **0** | **0** | **0** | **1** | **1** | **4** | **9** | **21** | **30** | **19** | **12** | **97** |

| $P_{rain}$ (mm) | 10 | 11 | 12 | 1 | 2 | 3 | 4 | 5 | 6 | 7 | 8 | 9 | year |
|---|---|---|---|---|---|---|---|---|---|---|---|---|---|
| 2001 | 1 | 0 | 0 | 0 | 0 | 1 | 3 | 3 | 7 | 10 | 13 | 8 | **45** |
| 2002 | 0 | 0 | 0 | 0 | 0 | 0 | 4 | 8 | 19 | 19 | 9 | 10 | **70** |
| 2003 | 0 | 0 | 0 | 0 | 0 | 1 | 7 | 5 | 8 | 12 | 15 | 3 | **52** |
| 2004 | 0 | 0 | 0 | 0 | 0 | 1 | 3 | 7 | 14 | 16 | 18 | 3 | **62** |
| 2005 | 0 | 0 | 0 | 0 | 1 | 1 | 2 | 8 | 15 | 42 | 28 | 12 | **110** |
| 2006 | 0 | 0 | 0 | 0 | 0 | 0 | 5 | 5 | 19 | 36 | 16 | 6 | **87** |
| 2007 | 2 | 1 | 0 | 0 | 0 | 1 | 5 | 5 | 31 | 26 | 24 | 16 | **110** |
| 2008 | 1 | 0 | 0 | 0 | 0 | 1 | 3 | 10 | 16 | 46 | 15 | 19 | **112** |
| 2009 | 3 | 0 | 0 | 0 | 0 | 1 | 3 | 13 | 11 | 44 | 23 | 27 | **126** |
| 2010 | 1 | 0 | 0 | 0 | 1 | 1 | 2 | 14 | 54 | 36 | 14 | 17 | **141** |
| 2011 | 1 | 0 | 0 | 0 | 1 | 1 | 3 | 10 | 33 | 22 | 30 | 12 | **113** |
| 2012 | 2 | 1 | 0 | 0 | 0 | 0 | 2 | 17 | 28 | 51 | 35 | 5 | **142** |
| 2013 | 2 | 0 | 0 | 0 | 0 | 1 | 2 | 11 | 18 | 33 | 16 | 9 | **93** |
| 2014 | 1 | 0 | 0 | 0 | 0 | 1 | 4 | 4 | 24 | 26 | 20 | 14 | **95** |
| **mean** | **1** | **0** | **0** | **0** | **1** | **1** | **4** | **9** | **21** | **30** | **19** | **12** | **97** |

95

**Table S5: Monthly and annual snowfall $P_{snow}$ (in mm/month or mm/a) in the Qaidam Basin (QB) during the 14 hydrological years (2001-2014) covered by the HAR 10 km data set** .

| $P_{snow}$ | 10 | 11 | 12 | 1 | 2 | 3 | 4 | 5 | 6 | 7 | 8 | 9 | year |
|---|---|---|---|---|---|---|---|---|---|---|---|---|---|
| 2001 | 4 | 4 | 6 | 3 | 3 | 2 | 16 | 13 | 10 | 1 | 7 | 7 | **77** |
| 2002 | 2 | 1 | 6 | 3 | 3 | 10 | 12 | 17 | 15 | 6 | 3 | 12 | **90** |
| 2003 | 5 | 3 | 6 | 4 | 8 | 16 | 21 | 18 | 11 | 9 | 11 | 7 | **119** |
| 2004 | 3 | 5 | 5 | 8 | 10 | 10 | 9 | 22 | 10 | 8 | 6 | 5 | **101** |
| 2005 | 6 | 4 | 8 | 3 | 6 | 12 | 7 | 19 | 14 | 13 | 11 | 14 | **119** |
| 2006 | 5 | 2 | 1 | 3 | 12 | 4 | 11 | 11 | 18 | 3 | 7 | 6 | **83** |
| 2007 | 6 | 2 | 2 | 1 | 5 | 13 | 9 | 15 | 28 | 9 | 4 | 17 | **112** |
| 2008 | 6 | 2 | 1 | 6 | 6 | 6 | 10 | 14 | 10 | 14 | 10 | 16 | **100** |
| 2009 | 10 | 5 | 6 | 6 | 9 | 13 | 8 | 31 | 9 | 13 | 10 | 11 | **130** |
| 2010 | 11 | 5 | 4 | 5 | 6 | 20 | 12 | 25 | 23 | 6 | 4 | 9 | **129** |
| 2011 | 10 | 1 | 3 | 3 | 5 | 11 | 11 | 25 | 16 | 7 | 5 | 10 | **108** |
| 2012 | 5 | 4 | 1 | 3 | 6 | 10 | 8 | 27 | 18 | 9 | 5 | 9 | **104** |
| 2013 | 10 | 4 | 4 | 3 | 7 | 1 | 5 | 19 | 6 | 8 | 2 | 14 | **81** |
| 2014 | 4 | 3 | 2 | 1 | 5 | 8 | 15 | 10 | 17 | 5 | 7 | 16 | **93** |
| **mean** | **6** | **3** | **4** | **4** | **6** | **10** | **11** | **19** | **15** | **8** | **7** | **11** | 103 |

100

**Table S6: Monthly and annual actual evapotranspiration**  **$ET$ (in mm/month or mm/a) in the Qaidam Basin (QB) during the 14 hydrological years (2001-2014) covered by the HAR 10 km data set**.

| ET | 10 | 11 | 12 | 1 | 2 | 3 | 4 | 5 | 6 | 7 | 8 | 9 | year |
|---|---|---|---|---|---|---|---|---|---|---|---|---|---|
| 2001 | 13 | 7 | 5 | 5 | 8 | 16 | 22 | 30 | 28 | 32 | 31 | 20 | **216** |
| 2002 | 10 | 5 | 4 | 4 | 8 | 14 | 18 | 26 | 33 | 34 | 27 | 20 | **204** |
| 2003 | 12 | 7 | 5 | 5 | 9 | 17 | 21 | 26 | 25 | 28 | 27 | 18 | **200** |
| 2004 | 10 | 6 | 4 | 5 | 8 | 14 | 17 | 25 | 25 | 30 | 32 | 17 | **193** |
| 2005 | 10 | 6 | 5 | 6 | 8 | 14 | 20 | 23 | 27 | 34 | 31 | 21 | **208** |
| 2006 | 11 | 6 | 4 | 6 | 7 | 10 | 13 | 19 | 27 | 34 | 25 | 17 | **179** |
| 2007 | 11 | 7 | 5 | 4 | 8 | 16 | 15 | 20 | 31 | 38 | 32 | 24 | **210** |
| 2008 | 17 | 8 | 6 | 5 | 7 | 13 | 18 | 22 | 30 | 35 | 37 | 23 | **222** |
| 2009 | 17 | 9 | 6 | 6 | 9 | 16 | 18 | 27 | 29 | 37 | 32 | 26 | **233** |
| 2010 | 16 | 10 | 6 | 7 | 9 | 17 | 20 | 23 | 36 | 39 | 30 | 20 | **234** |
| 2011 | 18 | 10 | 7 | 6 | 11 | 16 | 18 | 28 | 33 | 34 | 34 | 23 | **238** |
| 2012 | 16 | 10 | 5 | 4 | 9 | 17 | 20 | 32 | 32 | 34 | 36 | 21 | **238** |
| 2013 | 13 | 8 | 5 | 5 | 9 | 11 | 12 | 29 | 29 | 35 | 30 | 22 | **209** |
| 2014 | 14 | 8 | 4 | 6 | 8 | 13 | 24 | 21 | 24 | 34 | 31 | 23 | **212** |
| mean | **13** | **8** | **5** | **5** | **9** | **15** | **18** | **25** | **29** | **34** | **31** | **21** | **214** |

105

| ET (mm) | 10 | 11 | 12 | 1 | 2 | 3 | 4 | 5 | 6 | 7 | 8 | 9 | year |
|---|---|---|---|---|---|---|---|---|---|---|---|---|---|
| 2001 | 13 | 7 | 5 | 5 | 8 | 16 | 22 | 30 | 28 | 32 | 31 | 20 | **216** |
| 2002 | 10 | 5 | 4 | 4 | 8 | 14 | 18 | 26 | 33 | 34 | 27 | 20 | **204** |
| 2003 | 12 | 7 | 5 | 5 | 9 | 17 | 21 | 26 | 25 | 28 | 27 | 18 | **200** |
| 2004 | 10 | 6 | 4 | 5 | 8 | 14 | 17 | 25 | 25 | 30 | 32 | 17 | **193** |
| 2005 | 10 | 6 | 5 | 6 | 8 | 14 | 20 | 23 | 27 | 34 | 31 | 21 | **208** |
| 2006 | 11 | 6 | 4 | 6 | 7 | 10 | 13 | 19 | 27 | 34 | 25 | 17 | **179** |
| 2007 | 11 | 7 | 5 | 4 | 8 | 16 | 15 | 20 | 31 | 38 | 32 | 24 | **210** |
| 2008 | 17 | 8 | 6 | 5 | 7 | 13 | 18 | 22 | 30 | 35 | 37 | 23 | **222** |
| 2009 | 17 | 9 | 6 | 6 | 9 | 16 | 18 | 27 | 29 | 37 | 32 | 26 | **233** |
| 2010 | 16 | 10 | 6 | 7 | 9 | 17 | 20 | 23 | 36 | 39 | 30 | 20 | **234** |
| 2011 | 18 | 10 | 7 | 6 | 11 | 16 | 18 | 28 | 33 | 34 | 34 | 23 | **238** |
| 2012 | 16 | 10 | 5 | 4 | 9 | 17 | 20 | 32 | 32 | 34 | 36 | 21 | **238** |
| 2013 | 13 | 8 | 5 | 5 | 9 | 11 | 12 | 29 | 29 | 35 | 30 | 22 | **209** |
| 2014 | 14 | 8 | 4 | 6 | 8 | 13 | 24 | 21 | 24 | 34 | 31 | 23 | **212** |
| mean | **13** | **8** | **5** | **5** | **9** | **15** | **18** | **25** | **29** | **34** | **31** | **21** | **214** |

**Table S7: Monthly and annual water balance** ΔS (in mm/month or mm/a) in the Qaidam Basin (QB) during the 14 hydrological years (2001-2014) covered by the HAR 10 km data set .

| ΔS | 10 | 11 | 12 | 1 | 2 | 3 | 4 | 5 | 6 | 7 | 8 | 9 | year |
|---|---|---|---|---|---|---|---|---|---|---|---|---|---|
| 2001 | -8 | -3 | 1 | -2 | -5 | -13 | -3 | -13 | -11 | -21 | -11 | -5 | **-94** |
| 2002 | -8 | -4 | 2 | -1 | -4 | -4 | -2 | -2 | 2 | -9 | -15 | 1 | **-44** |
| 2003 | -6 | -4 | 1 | -1 | -1 | 0 | 7 | -2 | -6 | -7 | -1 | -8 | **-29** |
| 2004 | -7 | -1 | 1 | 3 | 2 | -3 | -5 | 3 | -1 | -6 | -8 | -9 | **-30** |
| 2005 | -4 | -2 | 3 | -3 | -2 | -1 | -11 | 4 | 2 | 21 | 8 | 5 | **21** |
| 2006 | -6 | -4 | -2 | -2 | 5 | -6 | 3 | -3 | 10 | 5 | -3 | -5 | **-10** |
| 2007 | -3 | -4 | -3 | -4 | -3 | -2 | -2 | 0 | 29 | -2 | -4 | 10 | **11** |
| 2008 | -10 | -7 | -4 | 1 | -1 | -7 | -5 | 2 | -5 | 25 | -12 | 12 | **-10** |
| 2009 | -5 | -4 | -1 | 0 | 0 | -1 | -7 | 17 | -9 | 20 | 1 | 12 | **24** |
| 2010 | -4 | -5 | -3 | -2 | -2 | 4 | -6 | 16 | 42 | 3 | -13 | 6 | **36** |
| 2011 | -7 | -9 | -3 | -3 | -5 | -4 | -4 | 6 | 16 | -5 | 1 | -1 | **-17** |
| 2012 | -9 | -5 | -4 | -1 | -3 | -7 | -10 | 12 | 14 | 26 | 3 | -7 | **8** |
| 2013 | -1 | -4 | -2 | -2 | -2 | -9 | -5 | 1 | -5 | 6 | -12 | 1 | **-35** |
| 2014 | -9 | -6 | -3 | -4 | -3 | -4 | -5 | -6 | 17 | -3 | -4 | 7 | **-24** |
| **mean** | **-6** | **-4** | **-1** | **-2** | **-2** | **-4** | **-4** | **2** | **7** | **4** | **-5** | **1** | **-14** |
| *P-ET* (mm) | 10 | 11 | 12 | 1 | 2 | 3 | 4 | 5 | 6 | 7 | 8 | 9 | year |
| 2001 | -8 | -3 | 1 | -2 | -5 | -13 | -3 | -13 | -11 | -21 | -11 | -5 | **-94** |
| 2002 | -8 | -4 | 2 | -1 | -4 | -4 | -2 | -2 | 2 | -9 | -15 | 1 | **-44** |
| 2003 | -6 | -4 | 1 | -1 | -1 | 0 | 7 | -2 | -6 | -7 | -1 | -8 | **-29** |
| 2004 | -7 | -1 | 1 | 3 | 2 | -3 | -5 | 3 | -1 | -6 | -8 | -9 | **-30** |
| 2005 | -4 | -2 | 3 | -3 | -2 | -1 | -11 | 4 | 2 | 21 | 8 | 5 | **21** |
| 2006 | -6 | -4 | -2 | -2 | 5 | -6 | 3 | -3 | 10 | 5 | -3 | -5 | **-10** |
| 2007 | -3 | -4 | -3 | -4 | -3 | -2 | -2 | 0 | 29 | -2 | -4 | 10 | **11** |
| 2008 | -10 | -7 | -4 | 1 | -1 | -7 | -5 | 2 | -5 | 25 | -12 | 12 | **-10** |
| 2009 | -5 | -4 | -1 | 0 | 0 | -1 | -7 | 17 | -9 | 20 | 1 | 12 | **24** |
| 2010 | -4 | -5 | -3 | -2 | -2 | 4 | -6 | 16 | 42 | 3 | -13 | 6 | **36** |
| 2011 | -7 | -9 | -3 | -3 | -5 | -4 | -4 | 6 | 16 | -5 | 1 | -1 | **-17** |
| 2012 | -9 | -5 | -4 | -1 | -3 | -7 | -10 | 12 | 14 | 26 | 3 | -7 | **8** |
| 2013 | -1 | -4 | -2 | -2 | -2 | -9 | -5 | 1 | -5 | 6 | -12 | 1 | **-35** |
| 2014 | -9 | -6 | -3 | -4 | -3 | -4 | -5 | -6 | 17 | -3 | -4 | 7 | **-24** |
| **mean** | **-6** | **-4** | **-1** | **-2** | **-2** | **-4** | **-4** | **2** | **7** | **4** | **-5** | **1** | **-14** |

**Table S8: Annual actual evapotranspiration** *ET*  in the Qaidam Basin (QB) during the calendar years 2001 to 2011 covered by both the HAR  (10 km data set and the SEBS data as published in Jin et al. (2013).

| ET (mm/a) | HAR 10 km | SEBS | Diff. |
|-----------|-----------|------|-------|
| 2001 | 211 | 73 | 138 |
| 2002 | 208 | 74 | 134 |
| 2003 | 196 | 78 | 118 |
| 2004 | 194 | 85 | 110 |
| 2005 | 205 | 144 | 61 |
| 2006 | 181 | 123 | 58 |
| 2007 | 219 | 135 | 84 |
| 2008 | 222 | 145 | 77 |
| 2009 | 232 | 170 | 62 |
| 2010 | 236 | 169 | 67 |
| 2011 | 234 | 182 | 52 |
| 2001-2011 | 213 | 125 | 87 |
| 2001-2004 | 202 | 77 | 125 |
| **2005-2011** | **218** | **153** | **66** |

| ET (mm/a) | HAR V1 | SEBS | Diff. |
|-----------|--------|------|-------|
| 2001 | 211 | 73 | 138 |
| 2002 | 208 | 74 | 134 |
| 2003 | 196 | 78 | 118 |
| 2004 | 194 | 85 | 110 |
| 2005 | 205 | 144 | 61 |
| 2006 | 181 | 123 | 58 |
| 2007 | 219 | 135 | 84 |
| 2008 | 222 | 145 | 77 |
| 2009 | 232 | 170 | 62 |
| 2010 | 236 | 169 | 67 |
| 2011 | 234 | 182 | 52 |
| 2001-2011 | 213 | 125 | 87 |
| 2001-2004 | 202 | 77 | 125 |
| **2005-2011** | **218** | **153** | **66** |

115

**Table S9:** Mean monthly and annual air temperature *T* (in deg C), specific humidity *q* (in g/kg), precipitation *P* (in mm/month or mm/a), rainfall $P_{rain}$ (in mm/month or mm/a), snowfall $P_{snow}$ (in mm/month or mm/a), actual evapotranspiration *ET* (in mm/month or mm/a), and water balance $\Delta S = P - ET$ (in mm/month or mm/a) in the Qaidam Basin (QB) as in Table 1 but derived from the HAR 30 km data set; sigma: standard deviations of annual values for each quantity during the hydrological years 2001 to 2014.

| month | 10 | 11 | 12 | 1 | 2 | 3 | 4 | 5 | 6 | 7 | 8 | 9 | **year** | sigma |
|---|---|---|---|---|---|---|---|---|---|---|---|---|---|---|
| *T* | -0.9 | -8.6 | -13.2 | -14.8 | -10.6 | -5.9 | 0.1 | 5.2 | 9.8 | 12.3 | 11.5 | 6.7 | **-0.7** | 0.6 |
| *q* | 2.3 | 1.4 | 1.1 | 1.0 | 1.3 | 1.5 | 2.0 | 2.9 | 4.4 | 5.8 | 5.3 | 4.0 | **2.8** | 0.2 |
| *P* | 8 | 4 | 4 | 4 | 7 | 11 | 15 | 29 | 39 | 39 | 28 | 23 | **210** | 45 |
| $P_{rain}$ | 1 | 1 | 0 | 0 | 0 | 1 | 4 | 8 | 22 | 29 | 19 | 11 | **95** | 30 |
| $P_{snow}$ | 7 | 3 | 4 | 4 | 7 | 10 | 11 | 21 | 17 | 10 | 9 | 12 | **115** | 20 |
| *ET* | 13 | 8 | 5 | 6 | 9 | 14 | 18 | 24 | 28 | 32 | 29 | 20 | **206** | 16 |
| $\Delta S$ | -5 | -4 | -1 | -2 | -1 | -4 | -3 | 5 | 11 | 7 | -2 | 3 | **3** | 36 |

| month | 10 | 11 | 12 | 1 | 2 | 3 | 4 | 5 | 6 | 7 | 8 | 9 | **year** | sigma |
|---|---|---|---|---|---|---|---|---|---|---|---|---|---|---|
| *T* (deg C) | -0.9 | -8.6 | -13.2 | -14.8 | -10.6 | -5.9 | 0.1 | 5.2 | 9.8 | 12.3 | 11.5 | 6.7 | **-0.7** | 0.6 |
| *q* (g/kg) | 2.3 | 1.4 | 1.1 | 1.0 | 1.3 | 1.5 | 2.0 | 2.9 | 4.4 | 5.8 | 5.3 | 4.0 | **2.8** | 0.2 |
| *P* (mm) | 8 | 4 | 4 | 4 | 7 | 11 | 15 | 29 | 39 | 39 | 28 | 23 | **210** | 45 |
| $P_{rain}$ (mm) | 1 | 1 | 0 | 0 | 0 | 1 | 4 | 8 | 22 | 29 | 19 | 11 | **95** | 30 |
| $P_{snow}$ (mm) | 7 | 3 | 4 | 4 | 7 | 10 | 11 | 21 | 17 | 10 | 9 | 12 | **115** | 20 |
| *ET* (mm) | 13 | 8 | 5 | 6 | 9 | 14 | 18 | 24 | 28 | 32 | 29 | 20 | **206** | 16 |
| *P*-*ET* (mm) | -5 | -4 | -1 | -2 | -1 | -4 | -3 | 5 | 11 | 7 | -2 | 3 | **3** | 36 |

---

## Author Response (AR3)

**General remarks**

Thanks to the valuable comments of reviewer 2 and the request by the editor, I have carefully revised the revised manuscript as explained below.

**Replies to Reviewer 2**

The new version has been improved. I would like to suggest accepting this work if the author can address a major comment and several minor ones below.

Major comment:

I agree that the dependence of precipitation on the change in air temperature and humidity can be physical, as constrained by physical processes in WRF. My point is, however, how evaporation from a mega-lake may alter the dependence of dS (storage change) on meteorological conditions. Current HAR data has included lake evaporation, but the lake area is small and thus its total evaporation amount is small, so the relationship between dS and meteorological conditions is essentially for land. In the mid-Pliocene, the mega-lake area is about 59,000 km^2, as stated in the introduction, which is about 23% of the QB area (254,000 km^2). In such a situation, lake evaporation can be a dominant player of the basin water balance, and current dependence of dS on meteorological conditions in Table 2 is not applicable to the basin scale in the mid-Pliocene.

Reply: I completely agree with the reviewer and have thus stated in the manuscript that this feedback and other feedbacks like additional increase in precipitation due to increase in local recycling of water evaporated from much larger lakes have not been considered in the study, so far. Based on the suggestions of the reviewer (see below), I decided to add an analysis of this feedback to the study.

Changes: Study added and text improved (see below).

Nevertheless, considering it is applicable to basin land, I suggest the authors to give a semi-empirical analysis through lake water balance, as stated below.

According to water balance of a lake for a long time,

lake evaporation – lake precipitation = runoff into the lake

If we assume the runoff into the lake is approximately the order of magnitude of land water storage in a close basin, as in Table 2, then

Lake area*( lake evaporation – lake precipitation)= Runoff*(basin area – lake area).

We can derive Runoff = Lake area*( lake evaporation – lake precipitation) / (basin area – lake area).

Given basin area=254000km^2, and lake area=59,000km^2 for the mid-Pliocene,

Runoff = 0.3*( lake evaporation – lake precipitation)

Lake evaporation is constrained by energy, and current studies suggest its annual amount is about 800 ~1000m (Lazhu et al., 2016, JGR, doi:10.1002/2015JD024523.; Li et al., 2017, JGR, DOI: 10.1002/2016JD025027).

For the mid-Pliocene, if lake precipitation = 300 mm/year, we can estimate

35 Runoff=150~210 mm/year.

The derived runoff value is much greater than the estimated dS in Table 2. If lake precipitation is as high as 700 mm/year, the required runoff to sustain the mega-lake is about 30~60 mm/year, which is close to the dS value in Table 2.

If such a high precipitation over the lake is not possible under current climate, then a warmer climate than that assumed here may be needed.

40 I hope this kind of discussion is helpful to improve our understanding.

Reply: Many thanks for these highly valuable suggestions. I have written a semi-empirical model (R source code) to compute the effects of changes in lake extent and precipitation on actual evapotranspiration, net precipitation, runoff from land areas to lakes, water storage and associated lake-level changes in the entire Qaidam Basin. The source code was then applied for an analysis of this feedback mechanism.

45 Changes: R source code of the semi-empirical model and a new figure (Fig. S10) added to the supplement. Analysis including a new figure (Fig. 5) and a new table (Table 3) added to the discussion in Section 4.3. Text adapted and improved at several locations. Literature references added.

Minor comments:

50 P4, at the end of L101, add "this data is used in the discussion" or similar. I took a long time to see how it is used in section 3, but actually It is used in Section 4.

Reply: Yes, I agree that it would help the reader to get information where the data are used.

Changes: Text improved by adding the sentence "The data set is used for a discussion of error and uncertainties (see Section 4)." to this paragraph and also to the preceding paragraph on GSOD data.

55

P6 L163, please confirm the number of 0.25%. Is it not 25%?

Reply: The very (and non-significant) number of 0.25% was again verified by me, which means that air temperature alone has no statistically significant effect. Air temperature shows an effect only in combination with specific humidity.

Changes: None.

60

P7 L196: Whang et al.. Is this a typo?

Reply: Yes, many thanks!

Changes: Typo corrected (Wang et al., 2014).

65 P7 L199, it is not true "Tibet tend to warmer but dryer climate". Overall, the Qinghai-Tibet Plateau is getting warmer and wetter except its southern and eastern edges. See Yang et al. (2014, Global and Planetary Change, 112: 79–91).

Reply: The research article of Zhang et al. (2013b) referenced in the manuscripts writes in Section 4.1: "*By averaging the climate data of all the 46 weather stations in the Qinghai-Tibetan Plateau [Fig. 2(a)], we found that the annual mean temperature showed a significant increase trend from 2000 to 2009 (p-value < 0.05) at a rate of 0.095 °C yr$^{-1}$. The temperature in Tibet had a higher mean annual value (5.15°C) and a higher increase rate (0.115 °C yr$^{-1}$) than those in Qinghai (2.45 °C yr$^{-1}$ and 0.074 °C yr$^{-1}$, respectively). For the entire plateau, the spatially averaged annual total precipitation showed large temporal variations during the 10 years. Regionally, Qinghai had a significant positive trend (10.7 mm yr$^{-1}$) and Tibet had a significant negative trend (-11.3 mm yr$^{-1}$). Therefore, the climate from 2000 to 2009 became warmer and drier in Tibet but warmer and wetter in Qinghai.*". This is exactly what I referred to.

The review article of Yang et al. (2014) and further studies are not in general contradiction to Zhang et al. (2013b) but more spatially detailed and also referring to different time periods. Since this study is not focussing on spatial and temporal variability of climate trends, I did not add a lengthy discussion on this topic but make a short notice on regional precipitation trends in Tibet.

Changes: Sentence and literature references added to improve the text.

[revised manuscript text omitted]
 10 km data set in the Qaidam Basin (QB) for the calendar years 2001-2011 (left panel) and 2005-2011 (right panel). Dotted lines: mean annual values; solid lines: regression lines; light grey shades: confidence intervals; dashed lines: prediction intervals.**

[Figure]

**Figure S10: Panel a): present-day lake extent in the Qaidam Basin (QB) as represented in the HAR 10 km data set. Panels b) to i): illustrations of lake extents for different projections of equilibrium lake states using the present-day model topography of the HAR 10 km data set for accumulation of net precipitation in the QB and subsequent runoff originating from land in areas below equilibrium lake levels (marked in cyan). Equilibrium lake levels (cf. Table 3): panel b) 2654 m a.s.l. (one metre above lowest present-day level); panel c) maximum extent of the mega-lake system of approx. 59.000 km² as reported by Chen and Bowler (1986); 2786 m a.s.l.; panel d) 2672 m a.sl. ($dP_{QB}$ = 50 mm/a; $ET_{lake}$ = 1000 mm/a); panel e) 2674 m a.sl. ($dP_{QB}$ = 50 mm/a; $ET_{lake}$ = 800 mm/a); panel f) 2678 m a.sl. ($dP_{QB}$ = 50 mm/a; $ET_{lake}$ = 600 mm/a); panel g) 2688 m a.sl. ($dP_{QB}$ = 100 mm/a; $ET_{lake}$ = 1000 mm/a); panel h) 2698 m a.sl. ($dP_{QB}$ = 100 mm/a; $ET_{lake}$ = 800 mm/a); panel i) 2711 m a.sl. ($dP_{QB}$ = 100 mm/a; $ET_{lake}$ = 600 mm/a). Black line: boundary of the QB (Lehner and Grill, 2013). Blue: present-day lake extent (1000 km²) as represented in the HAR 10 km data set. Topographic shading is based on DEM data from the SRTM.**

[Figure]

**Figure S110: Annual precipitation (upper left panel), actual evapotranspiration (upper right panel), and water balance (lower panel) in the Qaidam Basin (QB) during the hydrological years 2001 to 2014 as in Fig. 2 but derived from the HAR 30 km data set. Upper left panel: light grey bars: annual snowfall; dark grey bars: annual rainfall. Dotted lines: mean annual values.**

65

[Figure]

70 **Figure S12: Water balance versus precipitation (upper left panel) and actual evapotranspiration (upper right panel); precipitation versus air temperature (middle left panel) and specific humidity (middle right panel); water balance versus air temperature (lower left panel) and specific humidity (lower right panel) in the Qaidam Basin (QB) during the hydrological years 2001 to 2014 as in Fig. 4 but derived from the HAR 30 km data set. Dotted lines: mean annual values; solid lines: regression lines; light grey shades: confidence intervals; dashed lines: prediction intervals.**

 **2 Supplementary Tables**

**Table S2: Monthly and annual air temperature *T* (in deg C) in the Qaidam Basin (QB) during the 14 hydrological years (2001-2014) covered by the HAR 10 km data set.**

| *T* | 10 | 11 | 12 | 1 | 2 | 3 | 4 | 5 | 6 | 7 | 8 | 9 | **year** |
|------|------|-------|-------|-------|-------|------|------|-----|------|------|------|-----|----------|
| 2001 | -2.0 | -10.3 | -14.2 | -16.1 | -11.5 | -7.7 | -1.8 | 3.5 | 9.4 | 13.3 | 10.6 | 6.5 | **-1.6** |
| 2002 | -1.5 | -8.3 | -14.5 | -16.7 | -10.8 | -6.6 | 1.0 | 4.1 | 9.9 | 12.4 | 11.6 | 5.1 | **-1.2** |
| 2003 | -3.1 | -9.5 | -12.3 | -13.2 | -10.2 | -5.9 | 0.2 | 3.9 | 9.2 | 11.2 | 10.6 | 6.1 | **-1.0** |
| 2004 | -1.2 | -7.4 | -12.7 | -14.9 | -11.6 | -4.3 | 1.4 | 4.1 | 8.4 | 11.2 | 10.3 | 5.3 | **-0.9** |
| 2005 | -2.0 | -10.6 | -11.8 | -13.1 | -10.4 | -4.8 | -0.5 | 5.1 | 9.6 | 11.6 | 10.9 | 7.3 | **-0.7** |
| 2006 | -0.9 | -8.7 | -13.4 | -11.5 | -8.5 | -6.6 | -0.7 | 5.1 | 9.7 | 13.0 | 12.9 | 7.7 | **-0.1** |
| 2007 | 0.8 | -7.3 | -13.7 | -16.2 | -10.0 | -5.8 | 0.9 | 7.0 | 9.4 | 11.5 | 12.0 | 6.6 | **-0.4** |
| 2008 | -0.6 | -7.2 | -11.7 | -14.8 | -14.9 | -5.4 | 0.1 | 7.1 | 10.0 | 11.7 | 9.8 | 7.2 | **-0.7** |
| 2009 | 0.3 | -7.3 | -11.0 | -12.8 | -8.4 | -5.0 | 2.6 | 5.3 | 10.2 | 12.1 | 10.7 | 8.0 | **0.4** |
| 2010 | -0.6 | -8.2 | -12.6 | -11.5 | -8.9 | -5.3 | -0.8 | 5.4 | 9.8 | 13.9 | 12.8 | 7.7 | **0.2** |
| 2011 | -0.2 | -8.1 | -14.3 | -16.5 | -8.8 | -7.7 | 0.8 | 6.4 | 10.7 | 12.9 | 12.8 | 7.7 | **-0.3** |
| 2012 | 0.9 | -5.6 | -12.3 | -17.9 | -11.9 | -6.8 | 0.1 | 6.4 | 10.7 | 12.7 | 12.6 | 7.6 | **-0.3** |
| 2013 | -1.0 | -9.1 | -13.7 | -15.2 | -9.6 | -2.5 | 1.3 | 6.3 | 11.5 | 12.9 | 13.8 | 7.4 | **0.2** |
| 2014 | 0.8 | -10.3 | -15.2 | -14.1 | -11.3 | -4.4 | -0.1 | 5.0 | 10.2 | 13.4 | 11.6 | 7.9 | **-0.5** |
| **mean** | **-0.7** | **-8.4** | **-13.1** | **-14.6** | **-10.5** | **-5.6** | **0.3** | **5.3** | **9.9** | **12.4** | **11.6** | **7.0** | **-0.5** |

 **Table S2: Monthly and annual specific humidity *q* (in g/kg) in the Qaidam Basin (QB) during the 14 hydrological years (2001-2014) covered by the HAR 10 km data set.**

| *q* | 10 | 11 | 12 | 1 | 2 | 3 | 4 | 5 | 6 | 7 | 8 | 9 | **year** |
|------|-----|-----|-----|-----|-----|-----|-----|-----|-----|-----|-----|-----|----------|
| 2001 | 1.9 | 1.2 | 1.0 | 0.8 | 0.9 | 1.0 | 1.8 | 2.3 | 3.1 | 4.2 | 4.7 | 3.8 | **2.2** |
| 2002 | 1.7 | 1.3 | 1.0 | 0.9 | 1.1 | 1.4 | 2.2 | 2.7 | 4.0 | 5.5 | 4.4 | 4.0 | **2.5** |
| 2003 | 2.0 | 1.4 | 1.3 | 1.1 | 1.4 | 1.7 | 2.4 | 2.5 | 3.5 | 4.0 | 4.3 | 3.1 | **2.4** |
| 2004 | 1.8 | 1.6 | 1.1 | 1.1 | 1.3 | 1.7 | 1.9 | 2.7 | 3.3 | 4.5 | 5.0 | 2.9 | **2.4** |
| 2005 | 2.0 | 1.4 | 1.3 | 1.1 | 1.2 | 1.9 | 1.9 | 2.8 | 4.2 | 6.2 | 5.7 | 4.0 | **2.8** |
| 2006 | 2.0 | 1.2 | 0.8 | 1.1 | 1.5 | 1.1 | 1.7 | 2.4 | 4.3 | 6.4 | 5.1 | 3.5 | **2.6** |
| 2007 | 2.2 | 1.5 | 1.0 | 0.7 | 1.2 | 1.7 | 2.0 | 2.5 | 4.4 | 5.5 | 5.9 | 4.3 | **2.8** |
| 2008 | 2.7 | 1.3 | 1.0 | 1.0 | 1.1 | 1.5 | 1.8 | 3.0 | 4.2 | 6.1 | 4.6 | 4.5 | **2.7** |
| 2009 | 2.6 | 1.6 | 1.2 | 1.1 | 1.3 | 1.5 | 2.0 | 2.9 | 4.1 | 6.1 | 4.8 | 5.3 | **2.9** |
| 2010 | 2.4 | 1.4 | 1.1 | 1.1 | 1.4 | 1.7 | 2.0 | 3.1 | 5.6 | 7.1 | 5.1 | 4.7 | **3.1** |
| 2011 | 2.6 | 1.3 | 0.9 | 0.7 | 1.3 | 1.3 | 2.0 | 3.2 | 4.8 | 5.4 | 5.6 | 4.0 | **2.8** |
| 2012 | 2.3 | 1.7 | 0.9 | 0.8 | 1.2 | 1.5 | 1.7 | 3.4 | 5.0 | 6.5 | 6.0 | 3.1 | **2.8** |
| 2013 | 1.9 | 1.1 | 0.9 | 0.9 | 1.2 | 1.2 | 1.6 | 3.2 | 4.5 | 5.9 | 5.2 | 3.3 | **2.6** |
| 2014 | 2.0 | 1.2 | 0.9 | 0.8 | 1.1 | 1.5 | 2.0 | 2.1 | 4.6 | 5.4 | 5.0 | 4.2 | **2.6** |
| **mean** | **2.2** | **1.4** | **1.0** | **0.9** | **1.2** | **1.5** | **1.9** | **2.8** | **4.3** | **5.6** | **5.1** | **3.9** | **2.7** |

**Table S3: Monthly and annual precipitation _P_ (in mm/month or mm/a) in the Qaidam Basin (QB) during the 14 hydrological years (2001-2014) covered by the HAR 10 km data set.**

| _P_ | 10 | 11 | 12 | 1 | 2 | 3 | 4 | 5 | 6 | 7 | 8 | 9 | year |
|---|---|---|---|---|---|---|---|---|---|---|---|---|---|
| 2001 | 5 | 4 | 6 | 3 | 3 | 3 | 19 | 16 | 17 | 11 | 20 | 15 | **122** |
| 2002 | 2 | 1 | 6 | 3 | 3 | 10 | 16 | 25 | 34 | 25 | 12 | 22 | **160** |
| 2003 | 5 | 3 | 6 | 4 | 8 | 17 | 28 | 23 | 19 | 21 | 26 | 10 | **171** |
| 2004 | 3 | 5 | 5 | 8 | 10 | 11 | 12 | 29 | 24 | 24 | 24 | 8 | **163** |
| 2005 | 6 | 4 | 8 | 3 | 7 | 13 | 9 | 27 | 29 | 55 | 39 | 26 | **229** |
| 2006 | 5 | 2 | 1 | 3 | 12 | 4 | 16 | 16 | 37 | 39 | 23 | 12 | **170** |
| 2007 | 8 | 3 | 2 | 1 | 5 | 14 | 14 | 20 | 59 | 35 | 28 | 33 | **222** |
| 2008 | 7 | 2 | 1 | 6 | 6 | 7 | 13 | 24 | 26 | 60 | 25 | 35 | **212** |
| 2009 | 13 | 5 | 6 | 6 | 9 | 14 | 11 | 44 | 20 | 57 | 33 | 38 | **256** |
| 2010 | 12 | 5 | 4 | 5 | 7 | 21 | 14 | 39 | 77 | 42 | 18 | 26 | **270** |
| 2011 | 11 | 1 | 3 | 3 | 6 | 12 | 14 | 35 | 49 | 29 | 35 | 22 | **221** |
| 2012 | 7 | 5 | 1 | 3 | 6 | 10 | 10 | 44 | 46 | 60 | 40 | 14 | **246** |
| 2013 | 12 | 4 | 4 | 3 | 7 | 2 | 7 | 30 | 24 | 41 | 18 | 23 | **174** |
| 2014 | 5 | 3 | 2 | 1 | 5 | 9 | 19 | 14 | 41 | 31 | 27 | 30 | **188** |
| mean | **7** | **3** | **4** | **4** | **7** | **11** | **15** | **28** | **36** | **38** | **26** | **23** | **200** |

85

**Table S4: Monthly and annual rainfall $P_{rain}$ (in mm/month or mm/a) in the Qaidam Basin (QB) during the 14 hydrological years (2001-2014) covered by the HAR 10 km data set.**

| $P_{rain}$ | 10 | 11 | 12 | 1 | 2 | 3 | 4 | 5 | 6 | 7 | 8 | 9 | year |
|---|---|---|---|---|---|---|---|---|---|---|---|---|---|
| 2001 | 1 | 0 | 0 | 0 | 0 | 1 | 3 | 3 | 7 | 10 | 13 | 8 | **45** |
| 2002 | 0 | 0 | 0 | 0 | 0 | 0 | 4 | 8 | 19 | 19 | 9 | 10 | **70** |
| 2003 | 0 | 0 | 0 | 0 | 0 | 1 | 7 | 5 | 8 | 12 | 15 | 3 | **52** |
| 2004 | 0 | 0 | 0 | 0 | 0 | 1 | 3 | 7 | 14 | 16 | 18 | 3 | **62** |
| 2005 | 0 | 0 | 0 | 0 | 1 | 1 | 2 | 8 | 15 | 42 | 28 | 12 | **110** |
| 2006 | 0 | 0 | 0 | 0 | 0 | 0 | 5 | 5 | 19 | 36 | 16 | 6 | **87** |
| 2007 | 2 | 1 | 0 | 0 | 0 | 1 | 5 | 5 | 31 | 26 | 24 | 16 | **110** |
| 2008 | 1 | 0 | 0 | 0 | 0 | 1 | 3 | 10 | 16 | 46 | 15 | 19 | **112** |
| 2009 | 3 | 0 | 0 | 0 | 0 | 1 | 3 | 13 | 11 | 44 | 23 | 27 | **126** |
| 2010 | 1 | 0 | 0 | 0 | 1 | 1 | 2 | 14 | 54 | 36 | 14 | 17 | **141** |
| 2011 | 1 | 0 | 0 | 0 | 1 | 1 | 3 | 10 | 33 | 22 | 30 | 12 | **113** |
| 2012 | 2 | 1 | 0 | 0 | 0 | 0 | 2 | 17 | 28 | 51 | 35 | 5 | **142** |
| 2013 | 2 | 0 | 0 | 0 | 0 | 1 | 2 | 11 | 18 | 33 | 16 | 9 | **93** |
| 2014 | 1 | 0 | 0 | 0 | 0 | 1 | 4 | 4 | 24 | 26 | 20 | 14 | **95** |
| mean | **1** | **0** | **0** | **0** | **1** | **1** | **4** | **9** | **21** | **30** | **19** | **12** | **97** |

**Table S5: Monthly and annual snowfall $P_{snow}$ (in mm/month or mm/a) in the Qaidam Basin (QB) during the 14 hydrological years (2001-2014) covered by the HAR 10 km data set.**

| $P_{snow}$ | 10 | 11 | 12 | 1 | 2 | 3 | 4 | 5 | 6 | 7 | 8 | 9 | year |
|---|---|---|---|---|---|---|---|---|---|---|---|---|---|
| 2001 | 4 | 4 | 6 | 3 | 3 | 2 | 16 | 13 | 10 | 1 | 7 | 7 | **77** |
| 2002 | 2 | 1 | 6 | 3 | 3 | 10 | 12 | 17 | 15 | 6 | 3 | 12 | **90** |
| 2003 | 5 | 3 | 6 | 4 | 8 | 16 | 21 | 18 | 11 | 9 | 11 | 7 | **119** |
| 2004 | 3 | 5 | 5 | 8 | 10 | 10 | 9 | 22 | 10 | 8 | 6 | 5 | **101** |
| 2005 | 6 | 4 | 8 | 3 | 6 | 12 | 7 | 19 | 14 | 13 | 11 | 14 | **119** |
| 2006 | 5 | 2 | 1 | 3 | 12 | 4 | 11 | 11 | 18 | 3 | 7 | 6 | **83** |
| 2007 | 6 | 2 | 2 | 1 | 5 | 13 | 9 | 15 | 28 | 9 | 4 | 17 | **112** |
| 2008 | 6 | 2 | 1 | 6 | 6 | 6 | 10 | 14 | 10 | 14 | 10 | 16 | **100** |
| 2009 | 10 | 5 | 6 | 6 | 9 | 13 | 8 | 31 | 9 | 13 | 10 | 11 | **130** |
| 2010 | 11 | 5 | 4 | 5 | 6 | 20 | 12 | 25 | 23 | 6 | 4 | 9 | **129** |
| 2011 | 10 | 1 | 3 | 3 | 5 | 11 | 11 | 25 | 16 | 7 | 5 | 10 | **108** |
| 2012 | 5 | 4 | 1 | 3 | 6 | 10 | 8 | 27 | 18 | 9 | 5 | 9 | **104** |
| 2013 | 10 | 4 | 4 | 3 | 7 | 1 | 5 | 19 | 6 | 8 | 2 | 14 | **81** |
| 2014 | 4 | 3 | 2 | 1 | 5 | 8 | 15 | 10 | 17 | 5 | 7 | 16 | **93** |
| **mean** | **6** | **3** | **4** | **4** | **6** | **10** | **11** | **19** | **15** | **8** | **7** | **11** | **103** |

**Table S6: Monthly and annual actual evapotranspiration $ET$ (in mm/month or mm/a) in the Qaidam Basin (QB) during the 14 hydrological years (2001-2014) covered by the HAR 10 km data set.**

| $ET$ | 10 | 11 | 12 | 1 | 2 | 3 | 4 | 5 | 6 | 7 | 8 | 9 | year |
|---|---|---|---|---|---|---|---|---|---|---|---|---|---|
| 2001 | 13 | 7 | 5 | 5 | 8 | 16 | 22 | 30 | 28 | 32 | 31 | 20 | **216** |
| 2002 | 10 | 5 | 4 | 4 | 8 | 14 | 18 | 26 | 33 | 34 | 27 | 20 | **204** |
| 2003 | 12 | 7 | 5 | 5 | 9 | 17 | 21 | 26 | 25 | 28 | 27 | 18 | **200** |
| 2004 | 10 | 6 | 4 | 5 | 8 | 14 | 17 | 25 | 25 | 30 | 32 | 17 | **193** |
| 2005 | 10 | 6 | 5 | 6 | 8 | 14 | 20 | 23 | 27 | 34 | 31 | 21 | **208** |
| 2006 | 11 | 6 | 4 | 6 | 7 | 10 | 13 | 19 | 27 | 34 | 25 | 17 | **179** |
| 2007 | 11 | 7 | 5 | 4 | 8 | 16 | 15 | 20 | 31 | 38 | 32 | 24 | **210** |
| 2008 | 17 | 8 | 6 | 5 | 7 | 13 | 18 | 22 | 30 | 35 | 37 | 23 | **222** |
| 2009 | 17 | 9 | 6 | 6 | 9 | 16 | 18 | 27 | 29 | 37 | 32 | 26 | **233** |
| 2010 | 16 | 10 | 6 | 7 | 9 | 17 | 20 | 23 | 36 | 39 | 30 | 20 | **234** |
| 2011 | 18 | 10 | 7 | 6 | 11 | 16 | 18 | 28 | 33 | 34 | 34 | 23 | **238** |
| 2012 | 16 | 10 | 5 | 4 | 9 | 17 | 20 | 32 | 32 | 34 | 36 | 21 | **238** |
| 2013 | 13 | 8 | 5 | 5 | 9 | 11 | 12 | 29 | 29 | 35 | 30 | 22 | **209** |
| 2014 | 14 | 8 | 4 | 6 | 8 | 13 | 24 | 21 | 24 | 34 | 31 | 23 | **212** |
| **mean** | **13** | **8** | **5** | **5** | **9** | **15** | **18** | **25** | **29** | **34** | **31** | **21** | **214** |

**Table S7: Monthly and annual water balance ΔS (in mm/month or mm/a) in the Qaidam Basin (QB) during the 14 hydrological years (2001-2014) covered by the HAR 10 km data set.**

| ΔS | 10 | 11 | 12 | 1 | 2 | 3 | 4 | 5 | 6 | 7 | 8 | 9 | year |
|---|---|---|---|---|---|---|---|---|---|---|---|---|---|
| 2001 | -8 | -3 | 1 | -2 | -5 | -13 | -3 | -13 | -11 | -21 | -11 | -5 | **-94** |
| 2002 | -8 | -4 | 2 | -1 | -4 | -4 | -2 | -2 | 2 | -9 | -15 | 1 | **-44** |
| 2003 | -6 | -4 | 1 | -1 | -1 | 0 | 7 | -2 | -6 | -7 | -1 | -8 | **-29** |
| 2004 | -7 | -1 | 1 | 3 | 2 | -3 | -5 | 3 | -1 | -6 | -8 | -9 | **-30** |
| 2005 | -4 | -2 | 3 | -3 | -2 | -1 | -11 | 4 | 2 | 21 | 8 | 5 | **21** |
| 2006 | -6 | -4 | -2 | -2 | 5 | -6 | 3 | -3 | 10 | 5 | -3 | -5 | **-10** |
| 2007 | -3 | -4 | -3 | -4 | -3 | -2 | -2 | 0 | 29 | -2 | -4 | 10 | **11** |
| 2008 | -10 | -7 | -4 | 1 | -1 | -7 | -5 | 2 | -5 | 25 | -12 | 12 | **-10** |
| 2009 | -5 | -4 | -1 | 0 | 0 | -1 | -7 | 17 | -9 | 20 | 1 | 12 | **24** |
| 2010 | -4 | -5 | -3 | -2 | -2 | 4 | -6 | 16 | 42 | 3 | -13 | 6 | **36** |
| 2011 | -7 | -9 | -3 | -3 | -5 | -4 | -4 | 6 | 16 | -5 | 1 | -1 | **-17** |
| 2012 | -9 | -5 | -4 | -1 | -3 | -7 | -10 | 12 | 14 | 26 | 3 | -7 | **8** |
| 2013 | -1 | -4 | -2 | -2 | -2 | -9 | -5 | 1 | -5 | 6 | -12 | 1 | **-35** |
| 2014 | -9 | -6 | -3 | -4 | -3 | -4 | -5 | -6 | 17 | -3 | -4 | 7 | **-24** |
| **mean** | **-6** | **-4** | **-1** | **-2** | **-2** | **-4** | **-4** | **2** | **7** | **4** | **-5** | **1** | **-14** |

100

**Table S8: Annual actual evapotranspiration ET in the Qaidam Basin (QB) during the calendar years 2001 to 2011 covered by both the HAR 10 km data set and the SEBS data as published in Jin et al. (2013).**

| *ET* (mm/a) | HAR 10 km | SEBS | Diff. |
|---|---|---|---|
| 2001 | 211 | 73 | 138 |
| 2002 | 208 | 74 | 134 |
| 2003 | 196 | 78 | 118 |
| 2004 | 194 | 85 | 110 |
| 2005 | 205 | 144 | 61 |
| 2006 | 181 | 123 | 58 |
| 2007 | 219 | 135 | 84 |
| 2008 | 222 | 145 | 77 |
| 2009 | 232 | 170 | 62 |
| 2010 | 236 | 169 | 67 |
| 2011 | 234 | 182 | 52 |
| 2001-2011 | 213 | 125 | 87 |
| 2001-2004 | 202 | 77 | 125 |
| **2005-2011** | **218** | **153** | **66** |

**Table S9: Mean monthly and annual air temperature $T$ (in deg C), specific humidity $q$ (in g/kg), precipitation $P$ (in mm/month or mm/a), rainfall $P_{rain}$ (in mm/month or mm/a), snowfall $P_{snow}$ (in mm/month or mm/a), actual evapotranspiration $ET$ (in mm/month or mm/a), and water balance $\Delta S = P - ET$ (in mm/month or mm/a) in the Qaidam Basin (QB) as in Table 1 but derived from the HAR 30 km data set; sigma: standard deviations of annual values for each quantity during the hydrological years 2001 to 2014.**

| month | 10 | 11 | 12 | 1 | 2 | 3 | 4 | 5 | 6 | 7 | 8 | 9 | **year** | sigma |
|---|---|---|---|---|---|---|---|---|---|---|---|---|---|---|
| $T$ | -0.9 | -8.6 | -13.2 | -14.8 | -10.6 | -5.9 | 0.1 | 5.2 | 9.8 | 12.3 | 11.5 | 6.7 | **-0.7** | 0.6 |
| $q$ | 2.3 | 1.4 | 1.1 | 1.0 | 1.3 | 1.5 | 2.0 | 2.9 | 4.4 | 5.8 | 5.3 | 4.0 | **2.8** | 0.2 |
| $P$ | 8 | 4 | 4 | 4 | 7 | 11 | 15 | 29 | 39 | 39 | 28 | 23 | **210** | 45 |
| $P_{rain}$ | 1 | 1 | 0 | 0 | 0 | 1 | 4 | 8 | 22 | 29 | 19 | 11 | **95** | 30 |
| $P_{snow}$ | 7 | 3 | 4 | 4 | 7 | 10 | 11 | 21 | 17 | 10 | 9 | 12 | **115** | 20 |
| $ET$ | 13 | 8 | 5 | 6 | 9 | 14 | 18 | 24 | 28 | 32 | 29 | 20 | **206** | 16 |
| $\Delta S$ | -5 | -4 | -1 | -2 | -1 | -4 | -3 | 5 | 11 | 7 | -2 | 3 | **3** | 36 |

**3 R Source code of the semi-empirical model for estimation of water-balance components in the Qaidam Basin**

```
* * *
**Semi-empirical model for computation of water balance components in the**
**Qaidam Basin (QB) for different projections of lake extent (A.lake),**
**precipitation change in entire QB (dP.QB) with respect to present-day**
**precipitation (HAR 10 km), and mean rate of lake evaporation (ET.lake).**
#
**Regard: increased lake extent with high volumes of total lake evaporation**
**would lead to an additional increase in specific humidity, and thus enhance**
**local recycling of water. This feedback process is not considered in the**
**computations.**
#
**The R function QB.dS provides estimates used in the 2nd revison of the**
**manuscript "Scherer, D.: Survival of the Qaidam Mega-Lake System under**
**Mid-Pliocene Climates and its Restoration under Future Climates",**
**submitted to HESS in June 2020.**
#
**All quantitities are expressed as mean annual values. The following**
**abbreviations are used in the R code for naming of variables:**
#
**Quantities:**
**A:     area (extent) (km^2)**
**P:     precipitation (mm/a)**
**ET:    actual evapotranspiration (mm/a)**
**P.net: net precipitation (P - ET) (mm/a)**
**R:     runoff from land areas into lakes (mm/a)**
**dS:    change in water storage (water balance) (mm/a)**
#
**Subscripts:**
**QB:    Qaidam Basin (entire drainage basin)**
**PD:    present-day (as represented in the HAR 10 km data set)**
**nml:   no mega-lake projection (present-day lake extent as in HAR 10 km)**
**land:  land area of QB**
**lake:  lake area of QB**
**low:   low-altitude areas in the QB (z < 2.8 km a.s.l.)**
**gw:    groundwater**
#
**Results of the computations are returned in a R data frame.**
#
**Author:      Dieter Scherer, Technische Universitaet Berlin, Germany**
**Last update: 18.06.2020**
#
* * *
QB.dS <- function(A.lake, dP.QB, ET.lake) {

  #***********************************************************
  # Quantities regarded not to change significantly over time
  #***********************************************************

  A.QB <- 254000 # km^2; total area (A) of entire QB

  # Sensitivities from HAR 10 km analysis

  dET.dP.QB <- 0.2748 # sensitivity of ET to changes in P averaged over entire QB
  ddS.dP.QB <- 0.7252 # sensitivity of dS to changes in P averaged over entire QB

  #*************************************************
  # Quantities for present day (PD) from HAR 10 km
  #*************************************************

  A.lake.PD <- 1000 # km^2; present-day lake area (no mega-lake system)

  P.QB.PD   <-  200 # mm/a; P averaged over entire QB
  P.low.PD  <-   40 # mm/a; P averaged over lower altitudes (z < 2.8 km a.s.l.)

  ET.QB.PD  <-  214 # mm/a; ET averaged over entire QB

  dS.QB.PD  <- P.QB.PD - ET.QB.PD # mm/a; dS (= P.net) averaged over entire QB

  #********************
  # Projected changes
  #********************

  A.land <- A.QB - A.lake # km^2; land area of QB

  # Changes in P are assumed to uniformly take place at all altitudes

  P.QB     <- P.QB.PD  + dP.QB # mm/a; P averaged over entire QB
  P.low    <- P.low.PD + dP.QB # mm/a; P averaged over lower altitudes (< 2.8 km a.s.l.)
```

```r
        P.lake   <- P.low           # mm/a; mega-lake system forms at lower altitudes

        P.land   <- (P.QB*A.QB-P.lake*A.lake)/A.land # mm/a; P averaged over land area

      # Changes for projection for QB with lake extent for PD (no mega-lake system)

        P.QB.nml   <- P.QB                       # mm/a; P  averaged over entire QB
        dS.QB.nml <- dS.QB.PD + ddS.dP.QB*dP.QB # mm/a; dS averaged over entire QB
        ET.QB.nml <- P.QB.nml - dS.QB.nml        # mm/a; ET averaged over entire QB
       #ET.QB.nml <- ET.QB.PD + dET.dP.QB*dP.QB # mm/a; ET averaged over entire QB

      # Changes in ET, net P (P-ET) and dS

        ET.land <- (ET.QB.nml*A.QB-ET.lake*A.lake.PD)/A.land # mm/a; ET averaged over land area
        ET.QB   <- (ET.land*A.land+ET.lake*A.lake)/A.QB      # mm/a; ET averaged over entire QB

        P.net.lake <- P.lake - ET.lake          # mm/a; P.net averaged over mega-lake system
        P.net.land <- P.land - ET.land          # mm/a; P.net averaged over land area
        P.net.QB   <- P.QB   - ET.QB            # mm/a; P.net averaged over entire QB

        dS.QB      <- P.net.QB                  # mm/a; dS (= P.net) averaged over entire QB

      # Estimate change in (mega-)lake system

      if (P.net.land >= 0) {
        R.gw    <- 0                    # mm/a; no groundwater recharge (aquifers are considered to be filled)
        R.lake  <- P.net.land*A.land/A.lake # mm/a; runoff from land area into lakes
      } else {
        R.gw    <- P.net.land           # mm/a; all water losses are due to groundwater discharge
        R.lake  <- 0                    # mm/a; no runoff from land area into lakes
      }

        dS.lake <- P.net.lake + R.lake # mm/a; change in lake water storage

      # Return results as data frame

        results <- data.frame(A.QB, A.land, A.lake, dP.QB, P.QB, P.land, P.lake,
                              ET.QB, ET.land, ET.lake, P.net.land, P.net.lake,
                              dS.QB, R.lake, dS.lake)

      return(results)
    }
* * *
**Values used in the Qaidam Basin study**
* * *
**Projections for changes in precipitation**

dP.QB.1 <- 100 # mm/a; upper estimate from HAR analysis: projection 1
dP.QB.2 <-  50 # mm/a; lower estimate from HAR analysis: projection 2

**Projections for mean rates of lake evaporation**

ET.lake.a <-  600 # mm/a; lower estimate from HAR analysis (mean ET.lake in HAR: 650 mm/a): projection a
ET.lake.b <-  800 # mm/a; medium estimate from literature and HAR analysis: projection b
ET.lake.c <- 1000 # mm/a; upper estimate from HAR analysis: projection c

**Approximate equilibrium requirements for changes in precipitation and mean rates of lake evaporation**

dP.QB.mle.a   <- 210 # mm/a; required for sustaining maximum extent of mega-lake system (for ET.lake.a)
dP.QB.mle.b   <- 270 # mm/a; required for sustaining maximum extent of mega-lake system (for ET.lake.b)
dP.QB.mle.c   <- 330 # mm/a; required for sustaining maximum extent of mega-lake system (for ET.lake.c)

dP.QB.HAR     <- 19.3 # mm/a; required for sustaining HAR lake extent (identical for ET.lake.a, ET.lake.b, and ET.lake.c)

ET.lake.mle.1 <- 260 # mm/a; required for sustaining maximum extent of mega-lake system (for dP.QB.1)
ET.lake.mle.2 <- 110 # mm/a; required for sustaining maximum extent of mega-lake system (for dP.QB.2)

**Lake extent either taken from HAR data and literature, or computed as equilibrium values by the semi-empirical model**

A.lake.mle <- 59000 # km^2; maximum extent of mega-lake system from literature          (HAR10: z <= 2786 m, A = 59200 km^2)
A.lake.1a  <- 25773 # km^2; approximate sustainable mega-lake extent for projection 1a (HAR10: z <= 2711 m, A = 25900 km^2)
A.lake.1b  <- 19580 # km^2; approximate sustainable mega-lake extent for projection 1b (HAR10: z <= 2698 m, A = 19700 km^2)
A.lake.1c  <- 15864 # km^2; approximate sustainable mega-lake extent for projection 1c (HAR10: z <= 2688 m, A = 15800 km^2)
A.lake.2a  <- 10423 # km^2; approximate sustainable mega-lake extent for projection 2a (HAR10: z <= 2678 m, A = 10800 km^2)
A.lake.2b  <-  8067 # km^2; approximate sustainable mega-lake extent for projection 2b (HAR10: z <= 2674 m, A =  8500 km^2)
A.lake.2c  <-  6654 # km^2; approximate sustainable mega-lake extent for projection 2c (HAR10: z <= 2672 m, A =  6800 km^2)
A.lake.PD  <-  1046 # km^2; present-day lake extent from literature                     (HAR10: z <= 2654 m, A =  1100 km^2)
A.lake.HAR <-  1000 # km^2; present-day lake extent as in HAR 10 km                      (HAR10: z <= 2653 m, A =  1000 km^2)
* * *
**Example computation (as template for own computations)**
* * *
results <- QB.dS(A.lake=A.lake.1a, dP.QB=dP.QB.1, ET.lake=ET.lake.a)
```